# A computational account of multiple motives guiding context-dependent prosocial behavior

Claire Lugrin[1], Jie Hu[1,2]*, Christian C. Ruff[1,3,4]*

1 Zurich Center for Neuroeconomics (ZNE), Department of Economics, University of Zurich, Zurich, Switzerland, 2 Shanghai Key Laboratory of Mental Health and Psychological Crisis Intervention, School of Psychology and Cognitive Science, East China Normal University, Shanghai, China, 3 Faculty of Medicine, University of Zurich, Zurich, Switzerland, 4 URPP Adaptive Brain Circuits in Development and Learning (URPP AdaBD), University of Zurich, Zurich, Switzerland

* jhu@psy.ecnu.edu.cn (JH); christian.ruff@uzh.ch (CCR)

## Abstract

Prosocial behaviors play a pivotal role for human societies, shaping critical domains such as healthcare, education, taxation, and welfare. Despite the ubiquity of norms that prescribe prosocial actions, individuals do not consistently adhere to them and often behave selfishly, thereby harming the collective good. Interventions aimed at promoting prosociality would therefore be beneficial but are often ineffective due to a limited understanding of the various motives that govern prosocial behavior across different contexts. Here we present a computational and experimental framework to identify motives underlying individual prosocial choices and to characterize how these vary across contexts with changing social norms. Using a series of experiments in which 575 participants either judge the normative appropriateness of different pro-social actions or choose between prosocial and selfish actions themselves, we first show that while most individuals are consistent in their judgements about the appropriateness of behaviors, the actual prosocial behavior varies strongly across people. We used computational decision models to quantify the conflicting motives underlying judgements and prosocial choices, combined with a clustering approach to characterize different types of individuals whose judgements and choices reflect different motivational profiles. We identified four such types: *Unconditionally selfish* participants never behave prosocially, *Cost-sensitive* participants behave selfishly when prosocial actions are costly, *Efficiency-sensitive* participants choose actions that maximize total wealth, and *Harm-sensitive* participants prioritize avoiding harming others. When these four types of participants were exposed to different social environments where norms were judged or followed more or less by a group, they responded in fundamentally different ways to this change in others' behavior. Our approach helps us better understand the origins of heterogeneity in prosocial judgments and behaviors and may have implications for policy making and the design of behavioral interventions.

**Data availability statement:** The data sets generated during and analyzed for the current study, and the code reproducing the analysis are publicly available in the OSF repository at: https://osf.io/9fa72/.

**Funding:** C.C.R has received funding from the European Research Council (ERC) under the European Union's Horizon 2020 research and innovation programme (grant agreement no. 725355, ERC consolidator grant BRAINCODES). C.C.R. received funding from the University Research Priority Program 'Adaptive Brain Circuits in Development and Learning' (grant no. URPP AdaBD) at the University of Zurich and the Swiss National Science Foundation (grant no. 100019L-173248). J.H. received funding from the National Natural Science Foundation of China (grant no. 32400881). The funders had no role in study design, data collection and analysis, decision to publish, or preparation of the manuscript.

**Competing interests:** The authors have declared that no competing interests exist.

## Author summary

Prosocial behaviors are crucial for our societies and need to be encouraged. Yet not everyone behaves prosocially or responds positively to interventions aimed at increasing prosociality. Individual differences in these responses are not fully understood and may stem from diverse individual motivations. Characterizing these motivations, and how they can change, may therefore be crucial for promoting prosociality. Here we present an experimental and computational framework that provides key insights into the motives underlying prosocial behaviors. Applied to several experiments with over 500 participants, our framework characterizes the motivational setup of four different types of individuals who (do not) behave prosocially for different reasons, and who react differently to changes in others' norm compliance. We propose a unifying framework that allows researchers to better characterize the diversity and dynamics of prosocial motivations and that may help with the design of policies and behavioral interventions.

## Introduction

Prosocial behaviors – defined as actions that benefit others [1] – are essential for the smooth functioning of our societies. The capacity to consider our peers in our decisions and act in favor of even complete strangers [2] allows us to maintain societal structures built on mutual trust and cooperation [3–5]. Most societies have crystallized important prosocial behaviors into enforceable *social norms* that reflect widely shared standards of how individuals ought to behave [6,7]. These rules generally prescribe prosocial actions [8] in prototypical situations, such as acting in a fair way [9], reciprocating favors [10], or helping others [11].

Despite the widespread knowledge about these prosocial norms, not everyone acts prosocially in all situations. Instead, many people behave selfishly even when this harms others, for example by evading paying their taxes, hoarding vital resources, or stealing from others, even though they may act prosocially in other situations. Previous research has suggested that such individual and contextual differences in prosocial behavior may arise from differences in how individuals perceive which norms apply in a given situation [12–14], or from varying levels of motivation to follow the relevant norms in different contexts [15–17]. Therefore, a necessary step for promoting prosocial behaviors is to characterize what specific perceptual, cognitive and motivational processes underlie them, and how these can change across different contexts.

An important distinction for this aim is that between prescriptive and descriptive norms, which differ in nature and in their influence on behavior [18]. Prescriptive norms are defined as what a group believes is an appropriate behavior: Beliefs about what one ought to do. Descriptive norms are based on observations of others' actual behavior: What the majority chooses to do. These two types of norms can have different (but interacting) influences on behavior [6,19–23]. Observing others' actions,

for example, seems to have a potent effect on teenage drinking [24] or recycling [25], while exposing individuals to verbal information about what ought to be done has had promising outcomes on environmental conservation [20,26]. Aligned prescriptive and descriptive norms can enhance the effectiveness of interventions [27], while conflicting norms reduce it [21], with descriptive norms exerting a stronger influence than prescriptive norms [22]. Why this is the case, however, remains unclear, largely because we lack an understanding of the cognitive processes that underlie the perception of prescriptive versus descriptive norms, and of the specific motivations that drive individuals to follow these norms in a specific situation.

Previous research on these issues has mainly investigated the effect of different types of norms on group behavior, but has not focused on individual differences in sensitivity to the different types of information [28]. That is, while social norms are by definition shared across people in a given culture, it has been suggested that people may vary in how much they perceive the norm to apply in a specific context [12], and how much their actual behavior is consistent with this perceived norm [13,14]. However, the behavioral evidence for such links between norm perception and behavior is mixed. On the one hand, some research shows that measures of what behavior people judge to be appropriate predicts the actions of independent participants [29,30]. On the other hand, brain stimulation studies suggest that complying with a norm versus judging the morality of compliance relies on at least partially different processes [31,32]. It is therefore unclear whether participants use the same evaluation process when determining whether an action is appropriate in a given context, compared to when they must choose what to do themselves, which could explain why some interventions targeting norm perception have only minor effects on behavior [22].

Moreover, even when people agree on the (in)appropriateness of actions in a given situation, their actual actions may be affected by multiple motives that can conflict with each other. For example, although the egalitarian norm of splitting money equally between two people is widely accepted, other motives such as harm-aversion (i.e., preventing others from suffering drastic losses) or efficiency preferences (i.e., preventing a reduction of the total benefit across all individuals) can counteract inequality aversion and deter people from choosing equal monetary distributions in specific contexts [33–35]. Thus, even though people in a group may hold similar beliefs about what constitutes appropriate behavior (i.e., representation of a social norm), the motives determining whether they will take the corresponding prosocial actions in a specific situation can vary across individuals.

To comprehensively elucidate these possible individual differences in prosocial behavior, here we propose a computational and experimental framework that allows us to capture individual differences in the perception of social norms, prosocial actions, and responses to social environments with changing norms. In our framework, we consider several separate strands from the previous literature and present unified computational tools to study distinct aspects of behavior. First, we examined how the prescriptive norms – beliefs about what others deem to be appropriate behaviors [30,36,37], also referred to as injunctive norms [20] or normative expectations [6] – may change in line with different conflicting motives in specific situations. Second, we evaluated how these motives translate into individual actions [29] and characterized differences in the decision mechanisms underlying prosocial actions and the perception of prescriptive norms [31]. Finally, we characterized the motivational profiles of different individuals [38,39] and illustrated the power of this computational characterization for better understanding the complex patterns of changes in prosocial behavior in response to changes in norms [19,40,41]. Thus, our framework enables direct comparison of the relative strength and inter-relatedness of multiple motives underlying prosocial behavior, across individuals and choice situations. Note that the motives influencing behavior may be extrinsic (i.e., linked to external constraints such as sanction threats [42,43] or reputation concerns [44]) or intrinsic (private), influencing behavior even in the absence of those constraints. A good example of the latter is norm internalization [45,46], where individuals have adopted a norm as a moral value [47] and automatically follow it. In our framework, we focus only on intrinsic motives, creating situations without any reputation, reciprocity or sanction concerns, to characterize individual differences in private motivations to act prosocially [48].

More specifically, our framework focuses on four general motives that have been proposed to underlie prosocial actions. First, individuals may have some *baseline* preferences for prosocial actions [47] that are independent of the specific choice outcomes [49,50] or contexts. At the extremes of such *unconditional* motivation are people who always behave prosocially or selfishly, regardless of the specific situation [50]. Second, people may show *outcome-based* prosocial motivation and evaluate actions differently depending on the stakes (costs and consequences) of the decisions [51–53]. That is, people may weigh the trade-off between their own cost (or benefit) and others' benefit (or cost) when they make prosocial decisions [54,55]. These stakes can be imbalanced, in that the harm caused by selfish actions may outweigh their benefits. For example, a person stealing a wallet gains the money inside the wallet, while the victim loses the money and must in addition replace personal documents. Third, the specific nature of this *outcome-based* prosocial motivation may depend on different behavioral goals: Some individuals may be concerned with the *efficiency* [34,52,56] of different actions, favoring actions that maximize wealth regardless of who benefits from it. Others may instead focus on the harm caused by their actions and try to prevent bad outcomes for others, either actively (i.e., helping others out of adversity) or passively (i.e., avoiding actions that would harm others for personal benefit) [52,55,57]. Finally, individuals may show *context-dependent* prosocial motivation and may prefer prosocial actions only in some situations but not in others, even though these actions would have the same consequences. This follows from numerous findings that decisions producing the same outcome may be interpreted and implemented differently, depending on the language used to describe choices [58–60]. For example, the same prisoner's dilemma games framed as a "stock market" versus a "community" game can elicit less cooperation [59,61]. Framing the same decisions as active versus passive harm also has different effects on behavior and brain activations [60], showing that decisions producing the same outcome may be interpreted and implemented differently depending on the contexts. While all these prosocial motivations have been documented in the literature, they have rarely been studied within an integrated framework (but see [57] for a related attempt). It is therefore unclear how they (co-)vary within and across people and situations, which can now be studied with our proposed framework.

Importantly, prosocial behavior is not static but may evolve, in particular when individuals witness changes in others' behavior, such as how others act (descriptive norms) or what others judge to be appropriate (prescriptive norms). Measuring how different individuals respond to these two types of changes appears crucial for understanding how prosociality evolves and can be promoted. That both types of norms may influence behavior is already documented in the literature [19,37,62]. However, interventions focusing on descriptive versus prescriptive norms appear to have different (but interacting) influences on behavior [20–22], with some studies showing potent effects of prescriptive messages [63], others showing larger effects of descriptive interventions [22], and yet others showing no significant effects of either descriptive or prescriptive norm-based interventions [64]. Previous research has, however, largely focused on the average, group-level, response to each intervention, while individual differences remain poorly documented [28]. Our proposed framework allows us to characterize such individual differences and to compare how individuals with different motivational profiles change their behavior when exposed to environments in which the strenght of norm compliance varies.

Taken together, our framework comprises a set of tasks and computational models that allow researchers to systematically disentangle the effects of different motives on prosocial behavior. We validate this framework in a series of experiments, showing that it can systematically characterize individual differences in prosocial decisions and perceptions of corresponding prescriptive norms. We show that prosocial norm perception and behavior are guided differentially by theoretically-identified motives and that people can be classified based on the individual strengths of these motives into four different types who choose selfish or prosocial actions for different reasons. These distinct types adapt differently to changes in others' norm-related behavior, across social environments where observed descriptive or prescriptive norms encourage either prosocial or selfish actions. This illustrates how our computational framework

may be useful for understanding the possibilities and constraints of interventions aimed at promoting prosocial behavior.

## Results

To measure and compare individuals' perceptions of social norms and preferences for prosocial actions, we designed a judgment and an action task. Both tasks were based on situations in which players chose between a prosocial and a selfish action (Figs 1a and S1). Two types of players were involved: Players A (decision makers) and Players B (victims). Both collected points through an effortful time-estimation game (see Methods, S1 Text and S1 Fig). Subsequently, Player A could take actions affecting their and Player B's final payoffs in two different contexts (Fig 1a and 1b). In a "Destroying" context, they chose to either wipe out the points of B to get some bonus points (selfish action) or to forgo this bonus to preserve the points of B (prosocial action). In a "Helping" context, a lottery destroyed the points of B and A could either give up their bonus to help B recover these points (prosocial action) or take the bonus points (selfish action). In both contexts, A taking the bonus resulted in B receiving 0 points, whereas A foregoing the bonus resulted in both players keeping their points. To evaluate the effects of misbalanced cost and consequences associated with prosocial decisions, we systematically varied the payoffs of both players (points earned through the time estimation game) and the bonus offered to A (benefits associated with the selfish choice, Fig 1d). This mirrors the fact that in naturalistic settings, the costs and consequences of selfish actions are rarely matched.

We measured the perception of prosocial norms with a judgment task in which participants evaluated the appropriateness of fictive players A's actions (Fig 1c). Participants were incentivized to report what they believed the majority of other participants in the session would indicate [29], allowing us to measure the prescriptive norm in the group rather than the private standards of each participant. We recruited 75 participants (Experiment 1) to play 250–300 trials of this judgment task (facing all combinations of contexts, points of A, B, bonus, and decisions of A, Fig 1d). Confirming the existence of prescriptions for prosocial actions, participants rated prosocial choices on average as "socially appropriate" (scale transformed to range from 0 "very socially inappropriate" to 1 "very socially appropriate"; mean = 0.82, std = 0.11), with only small variations across participants and situations (S2a–S2d Fig, Mixed-effects ordinal regression of judgments: S1a Table). By contrast, selfish actions were rated as "socially inappropriate" (mean = 0.26 points, std = 0.15, S2e Fig). The perceived appropriateness of selfish actions varied across situations: Higher bonus values (gained by A when acting selfishly) were associated with more lenient judgments, whereas higher points of Player B (lost by B if A acts selfishly) or points of Player A (earned by A regardless of their actions) were associated with harsher judgments (Fig 1e, S1b Table: bonus, Points B, Points A, all $p < 0.001$). Confirming a significant difference in normative beliefs between the two contexts, participants rated Destroying as more inappropriate than Not-helping (S2f Fig, 7% difference, Mixed-effects ordinal regression of judgments: S1b Table: context $p < 0.001$). These results confirm that specific features of the choice situation affect the strength of the general prescriptions against selfish actions.

To evaluate individuals' prosocial decisions, we recruited 70 participants to participate in the action task as Player A and choose between prosocial and selfish actions (Experiment 2). 34 participants played as Player B. Both types of participants played the time estimation game to earn the points used for the action task (See Methods and S1 Text for details). At the end of the experiment, three trials were randomly selected and both players were paid according to Player A's decisions. Prosocial decisions were therefore costly for Player A and selfish decisions had real consequences for another person. Participants chose the prosocial option on average 33% of the time (std = 0.27, ranging from 0% to 100% of the trials: S2g Fig). The cost and consequences of prosocial decisions affected participants' actions: They acted more selfishly for increasing bonus values and more prosocially with increasing points of Player B (Fig 1f, S2 Table, Mixed-effects logistic regression of actions: bonus and Points B $p < 0.001$). Participants chose to not help slightly more often than they

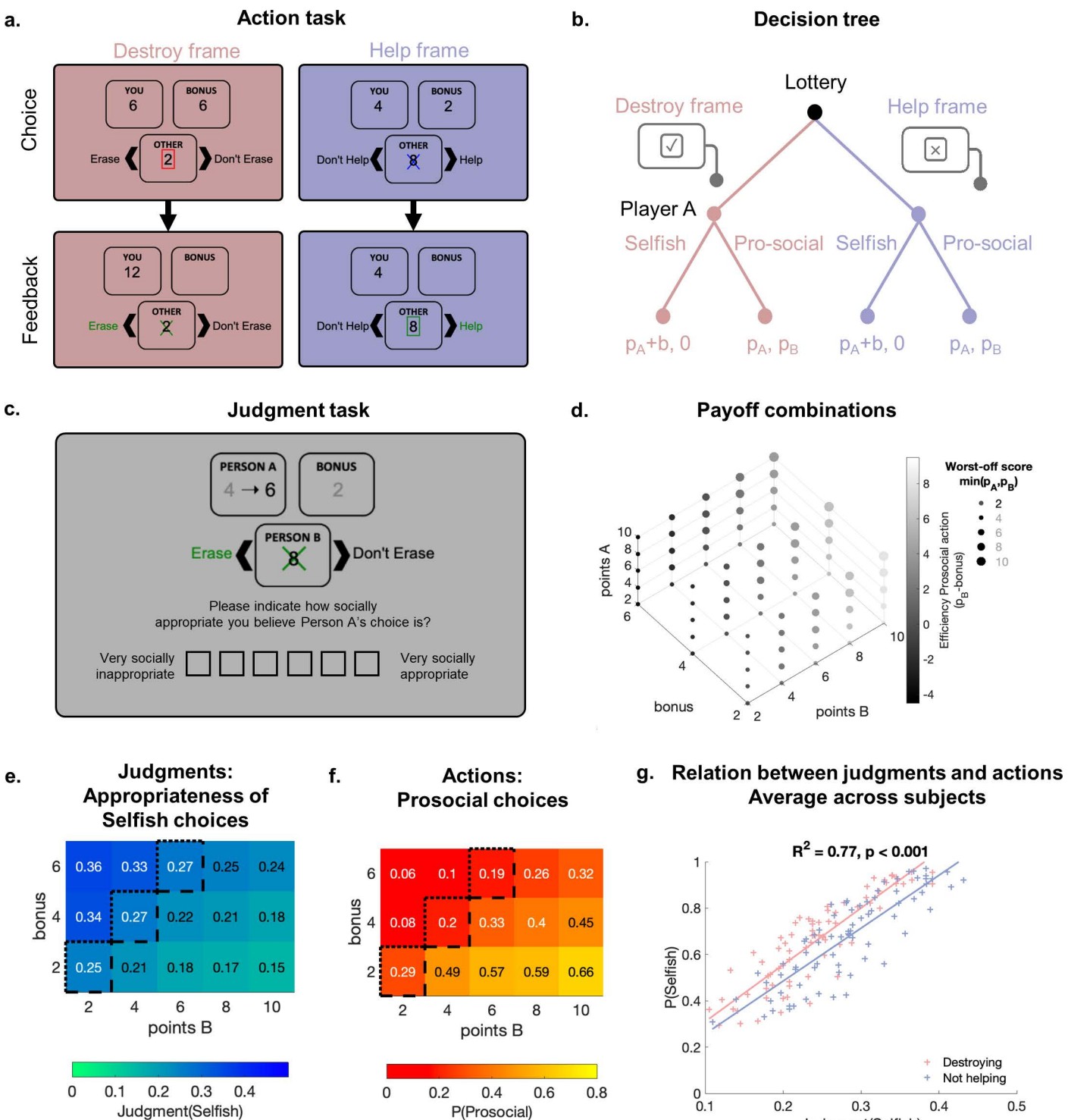

**Fig 1. Prosocial actions and the prescriptive norms associated with them vary across choice situations. (a)** Action task. Player A is shown a choice situation involving their outcome ($p_A$, labelled "you"), and the outcome of a second player ($p_B$, labelled "other") and chooses between a selfish and a prosocial action presented in one of two frames (Destroy in pink vs Help in purple). After they choose, they receive feedback on their decision's outcome. **(b)** Decision tree. The lottery determines the decision context (Destroy vs Help). For both contexts, Player A picking the selfish option results in them keeping their points ($p_A$) plus the bonus and Player B getting 0 points; Player A picking the prosocial option results in both Players A and B keeping

their points ($p_A$ and $p_B$). **(c)** Judgment task. Participants are shown fictive decisions of Player A and must rate the appropriateness of these decision on a 6-item Likert-scale from very socially inappropriate (1) to very socially appropriate (6). **(d)** Payoff combinations. The points earned by each player during the time-estimation task range from 2 to 10 and the bonus ranges from 2 to 6. All possible combinations of these 3 values are shown to the participants across trials. The grey scale represents the efficiency of prosocial decisions ($p_B$-bonus) and the bullet sizes the worst-off player score (min ($p_A$, $p_B$)). **(e-f)** Effect of payoff distribution (points of Player B and bonus) on average judgment (e), and prosocial action rate (f). The color scales represent the average behavior across participants of Experiment 1 (e) or 2 (f). The black lines represent efficiency thresholds: Selfish actions are efficient for trials above the dotted lines, and prosocial actions are efficient for trials below the dashed line. Trials between those two lines have equal efficiency for prosocial and selfish actions ($p_B$=b). **(g)** Relation between selfish actions rate averaged across participants of Experiment 2 and appropriateness judgments of selfish actions averaged across participants of Experiment 1, for the 75 different trials played in each context: Destroying in pink and Not Helping in purple. Each dot represents one trial averaged across all participants. The lines show regressions.

chose to actively destroy player B's points (2.5% difference, S2 Table: context p < 0.001). In line with the preceding judgment task, participants' actions depended on context and the consequences of the actions.

As reported in previous research studying judgments and actions at the group level [29], we found a strong correlation between the judgments averaged across all participants of Experiment 1 and the actions averaged across all participants of Experiment 2 on each trial (Fig 1g, $R^2$ = 0.77, p < 0.001). However, the variation of judgments and actions across choice situations highlights the complexity of such behaviors: There is not a single norm against destroying or in favor of helping, but different conflicting motives are at play in the different choice situations.

### Different motives guide actions versus judgments

To confirm that several distinct motives impact on the perception of prescriptive norms, we first turned to model-free analysis. This showed that judgments were affected by both the efficiency of prosocial choices (difference between the points of B preserved by the prosocial choice and the bonus lost by Player A) as well as the score of the worst-off player (minimum points collected by A and B; Experiment 1; Fig 2a, S3 Table, both p < 0.001). In contrast, prosocial actions were only significantly affected by the efficiency of these actions (Experiment 2, Fig 2b, S4 Table, efficiency p < 0.001), but not the score of the worst-off player (Fig 2b, S4 Table, Worst-off p = 0.14). This suggests that different motives may guide the perception of prescriptive prosocial norms compared to prosocial actions. Fig 2b additionally shows considerable differences between participants in the effects of efficiency and worst-off player points on prosocial actions, emphasizing the importance of characterizing these motives at the individual level.

To understand whether individual differences in prosocial actions may be linked to differences in perceptions of the norms, we conducted a new experiment in which participants played both the action and judgment tasks in a counterbalanced order, facing only "Destroying" trials (Experiment 3, N = 72). The findings of Experiments 1 and 2 largely replicated in this new sample (S3 Fig, S1 and S2 Tables, effect of Experiment p > 0.28 for all analyses). The order of the two tasks did not significantly affect the actions of the participants (S2 Table, task order p = 0.09) but did have an influence on their judgments (S1 Table, task order p = 0.01): Participants facing the judgment task first were slightly less lenient than those facing this task second. We again found that at the group level, average judgments predicted average actions across situations (S3g Fig, $R^2$ = 0.85, p < 0.001).

When focusing on the individual level, however, we surprisingly found that individual participant's judgments did not relate to their own actions: Judgment and action scores averaged across trials for each participant were not significantly correlated (Fig 2c, r = -0.15, p = 0.22). Average judgments were similar across participants (Fig 2c, mean = 0.28, std = 0.13), reflecting a consensus on normative judgments, but average prosocial action rates varied widely across participants (Fig 2c, mean = 0.47, std = 0.28). This shows that most participants agree on what behavior is prescribed, but not everyone follows these prescriptions.

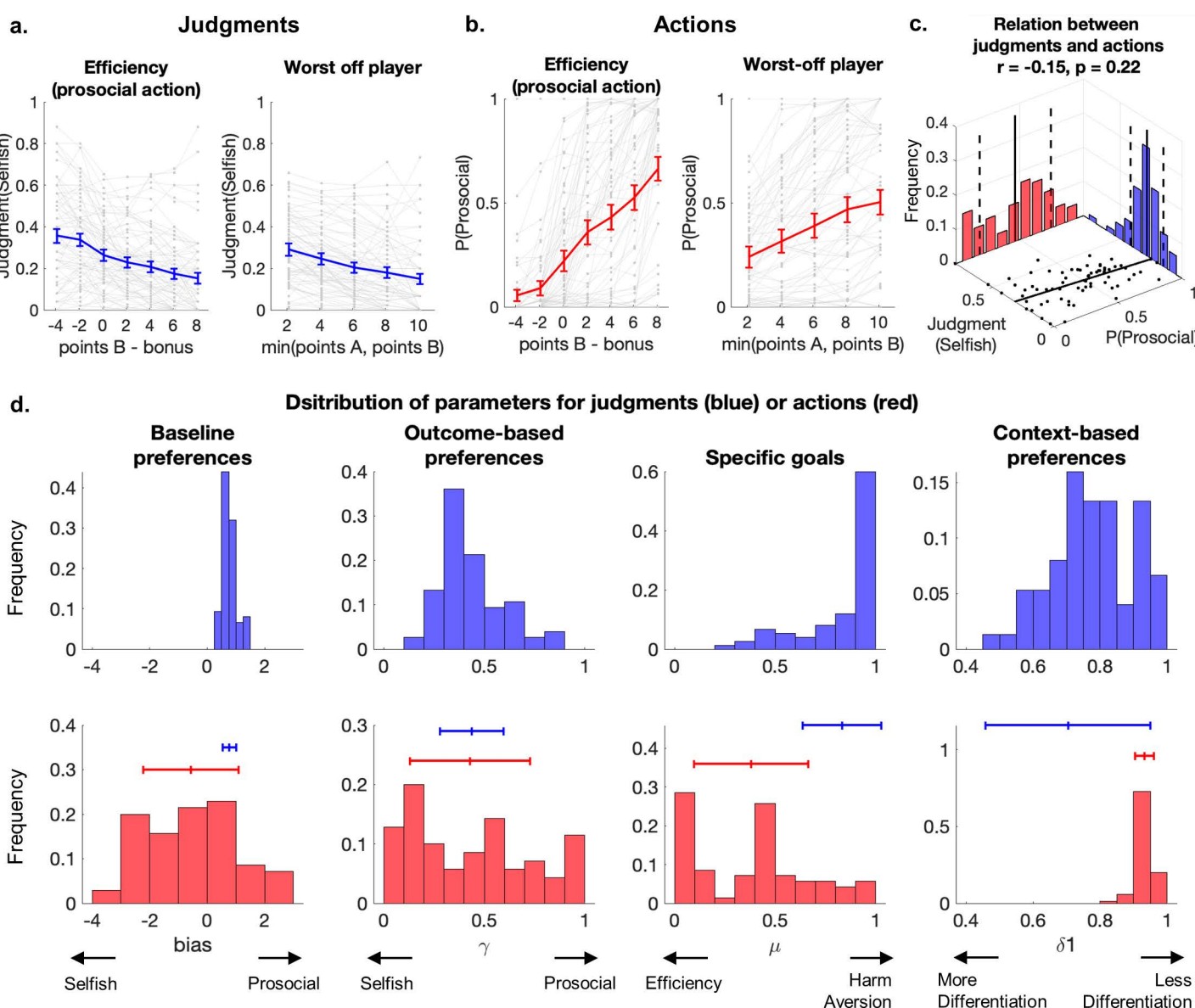

**Fig 2. Judgment and actions rely on different mechanisms that are captured by computational models. (a-b).** Relation between prosocial actions efficiency ($p_B$-b) or score of the worst-off player (min ($p_A$, $p_B$)) and average appropriateness judgments of participants of Experiment 1 **(a)** or frequency of prosocial actions for participants of Experiment 2 **(b)**. Each dot and grey line represent a participant, the colored lines are the average across participants, and error bars represent the standard error of the mean. **(c)** Bivariate distribution of the prosocial action rate (left) and mean appropriateness judgment of selfish choices (right), for each participant of Experiment 3. The means and standard deviation across participants are represented in plain (resp dotted) lines and show individual differences in actions but relative consensus in judgments. The black dots represent participants and show no significant correlation between the average actions and judgments. **(d)** Modeling of action and judgments. We used an adapted Charness and Rabin (CR $\delta 1$ bias) model to compute the utility of possible actions of Player A and to predict the trial-by-trial probability of prosocial action, or the probability of selecting each appropriateness rating. The histograms represent the participant-specific fitted parameters of the model for the judgment (blue) and action (red) tasks. *Baseline* preferences (bias) quantify the general tendency to act prosocially, independently of the stakes of the decision. The judgment bias parameter has been rescaled to yield responses between 0 (very socially appropriate) to 1 (very socially inappropriate), and thus in the same range as the binary actions (0: selfish; 1: prosocial). However, note that the bias parameter is an intercept that was allowed to take any value between negative infinite (extremely selfish) to positive infinite (extremely prosocial). *Outcome-based* preferences ($\gamma$) model the preference of each participant for prosocial versus selfish outcomes. *Specific goals* ($\mu$) quantify concerns for efficiency ($\mu = 0$) versus harm-aversion ($\mu = 1$). *Context-based* preferences ($\delta_1$) quantify the influence of the context (Destroying vs Helping). The red and blue lines represent the mean and standard deviations of the distributions, showing wider individual differences in parameters for the actions compared to judgments.

These results suggest that different motives guide the perceptions of prescriptive norms versus the associated prosocial actions. To quantify the effects of such concerns on the judgments and actions of distinct individuals, we used a set of computational utility models of social preferences and normative choices (See S5 Table for details of the different models). We compared models using a LOO approach [65], taking into account both the fit and the complexity of the models (S4 Fig and S6 and S7 Tables for a report of the expected log pointwise predictive density (elpd) reflecting out-of-sample predictive fit of the models). This showed that the participants' behavior was best captured by an extended version of the Charness and Rabin [52] social-preference model (CR $\delta_1$ bias). This model captured the difference in utility between prosocial and selfish actions ($\Delta U$) using four parameters: First, a bias term reflects *baseline preferences* for prosocial actions regardless of the stakes of the decisions. It is modeled as an added constant that influences all decisions alike and that can take theoretically any value from negative infinite (infinitely selfish) to positive infinite (infinitely prosocial). Second, an *outcome-based* term ($\gamma$) captures the extent to which participants value prosocial ($\gamma = 1$) versus selfish ($\gamma = 0$) outcomes. Third, a parameter $\mu$ captures the *specific goals* pursued by outcome-sensitive participants, with the assumption that such goals belong to a spectrum ranging from a pure goal of *efficiency* (maximizing the total payoff, $\mu = 0$) to a pure goal of *harm-aversion* (minimizing the loss of the worst-off player $\mu = 1$). Fourth, a discount factor ($\delta_i$) differentiates the two contexts (Not Destroying: $\delta_1$ and Helping: $\delta_2$), by discounting the bonus for the destroying context (No discount: $\delta_1 = 1$, full discount: $\delta_1 = 0$; $\delta_2$ fixed to 1 for parsimony). For the resulting CR $\delta1$ bias model, $\Delta U$ was computed as follows:

$$\Delta U \;=\; bias + \; \gamma\mu \min\left(p_A, p_B\right) + \; \gamma(1-\mu)\left[p_B - \delta_i b\right] - (1-\gamma)\delta_i b$$

where $p_A$ and $p_B$ are the points of Player A and B and $b$ the bonus value.

We used this utility difference to estimate the probability of selecting each appropriateness rating using an ordered-probit model, or the probability of choosing the prosocial action on a given trial using a SoftMax model (S1 Text), therefore testing the assumption that the same motives (parameters weighing the different possible payoffs) underlie both actions and judgments. We validated this assumption by using model comparison and model fitting techniques, showing that the model was able to capture participants' behavior. Simulations of behavior using fitted models illustrate how the CR $\delta1$ model faithfully predicts the average participants' behavior regardless of the choice situation (S4 Fig), while flexibly capturing the behavioral patterns of different participants (S5 Fig). Crucially, no parameters of the winning model were correlated, and all parameters recovered well (S6 Fig), confirming the importance and independence of all four motives for fully capturing the participant's decision mechanisms.

Fitting the model to the participants' judgments and actions allowed us to quantify the influence of each motive on these two types of decisions for each participant. We first confirmed that all fitted parameters fell within sensible ranges, reflecting some nuanced combinations of different motives. All parameters significantly influenced participants' actions and judgments (Linear regression of average actions or judgments and model parameters: S8 Table, all p<0.001), but to different degrees (Fig 2d). Most participants relied on similar motives when reporting their perceptions of prescriptive norms: They showed average *baseline* and *outcome-based* preferences, favored *harm-aversion* over *efficiency* goals (Fig 2d and S3 Table), and were affected by decision *contexts* (however, individual differences were evident for this parameter; Fig 2d). By contrast, participants of the action task were mostly influenced by *efficiency* concerns rather than *harm-aversion* (Fig 2d), showed strong individual differences in *outcome-based* and *baseline* preferences, and were hardly affected by decision *context*. Within-participant analysis of Experiment 3 showed low correlations between the fitted parameters for the action and judgment tasks (S3h Fig), and two-sample Kolmogorov-Smirnov tests confirmed that the distributions of the *outcome-based* ($\gamma$), *specific-goal* ($\mu$) and *context-based* ($\delta_1$) parameters were significantly different for the action versus judgment models (p<0.001, test statistic>0.34 for $\gamma$, $\mu$, and $\delta_1$; the bias distributions are not directly statistically comparable as they model data in different ranges – binary versus ordinal). Thus, these results highlight again a disconnect between individuals' understanding of how they should act and their actual prosocial behavior.

**Fig 3. Clustering of participants based on fitted action model parameters reveals four types of participants. (a)** Parameter combinations of each participant of Experiments 1 and 3, and centroids of the 4 clusters (large markers). Four different types of participants are represented by the different marker types and colors. **(b)** Number of participants of each type across Experiments 1 and 3. **(c)** Mean and standard error of the normalized parameters of the 4 types for participants of Experiments 1 and 3. The + and − signs indicate the parameters of interest for the different clusters. **(d)** Effects of bonus and points of player B on prosocial actions (top row, red) and judgments (bottom row, blue) averaged across each type of participant of

Experiment 3. The colors represent the prosocial action rate and the average judgment of selfish actions. The black lines represent efficiency thresholds: Selfish actions are efficient for trials above the dotted lines, and prosocial actions are efficient for trials below the dashed line.

## Four types of individuals with distinct patterns of motives

As individual differences in prosociality could not be attributed to differences in the perception of prescriptive norms, we focused the rest of our analysis on the four motives underlying actions captured by the CR bias model. The continuous variations in the fitted parameters capturing these motives (Fig 2d) suggest that a unique combination of motives underlie the behavior of each participant. To structure these individual differences and understand how they explain differences in prosocial behavior, we defined distinct types of individuals – or motivational profiles – by applying a clustering procedure (k-means) using the fitted parameters obtained from Experiments 1 and 3. Optimizing the number of clusters revealed four types of participants (Fig 3a–3c) whose actions differed significantly (ANOVA: F = 25.92, p < 0.001). A first type of participants, representing 20% of our sample, was *Unconditionally selfish*: They had a strong bias to act selfishly, regardless of the situation (Figs 3c and S7). A second cluster was composed of *Cost-sensitive prosocials* (25% of our sample): They had a high bias towards prosocial actions but also valued their own outcome (low $\gamma$). That is, they were generally prosocial but increasingly chose selfish actions when the bonus increased (Fig 3d). The remaining participants showed a high general concern for prosocial outcomes (high $\gamma$), but clustered into two types that differed in the nature of their *outcome-based* preferences: *Efficiency-sensitive prosocials* (39% of participants) favored *efficiency* over *harm-aversion* (low $\mu$) and picked the selfish action when the bonus value (cost of prosocial choices) was higher than the points of player B (consequences of selfish choices). *Harm-sensitive prosocials* (16% of participants) were the most prosocial, valued *harm-aversion* over *efficiency* (high $\mu$) and picked the selfish action only when the harm caused to the other player was low.

While the four different types of participants varied markedly in their choices across situations, they showed comparable judgments (Fig 3d; Effect of clusters on average judgment, ANOVA: F = 0.54, p = 0.66). This confirms that the participants' understanding of the prescriptive norm was independent of their motivational profiles.

We conducted an exploratory analysis of the ecological validity of the different motives and motivational profiles, by relating them to established subclinical personality traits (S9 Table and S8 Fig). *Outcome-based* and *baseline* preferences related to traits such as Empathic concern, Psychopathy, or Machiavellianism (p < 0.01, S8 Fig). Participants labeled as *Unconditionally selfish* had significantly higher Machiavellianism and primary psychopathy scores, and lower empathic concerns, than the rest of the participants (S8i Fig: ANOVA of the effects of clusters on scores: all p < 0.02, post-hoc two-tailed t-tests of cluster-specific differences: all p < 0.002). Thus, our exploratory analysis suggests that estimating model parameters and clustering participants may be useful for relating disorders of social behaviors to the different mechanisms captured by our model.

## Participants can adapt to prosocial and selfish environments

One crucial motivation behind studies of prosocial behavior is to understand how it can change following behavioral interventions [48,66], and what may predispose individuals to react differently. We therefore tested how the four types of individuals (identified in a replication sample with new participants, Experiment 4, N = 358) adapted their behavior after exposure to prosocial or selfish environments. Acknowledging the difference between descriptive and prescriptive norms, we exposed participants to two types of environments with different normative behavior (Fig 4a): one emphasizing a descriptive behavior (what others choose to do) and the other a prescriptive behavior (what others judge to be appropriate). For both environments, we investigated reactions to changes in a positive (encouraging prosocial actions) versus negative (encouraging selfish actions) direction (Fig 4b). We measured participants' actions and judgments before and after exposing them to situations of the action (resp. judgment) task. During the exposure, participants had to guess how many of 11 previous participants picked the selfish action (resp. what the most popular appropriateness judgment among

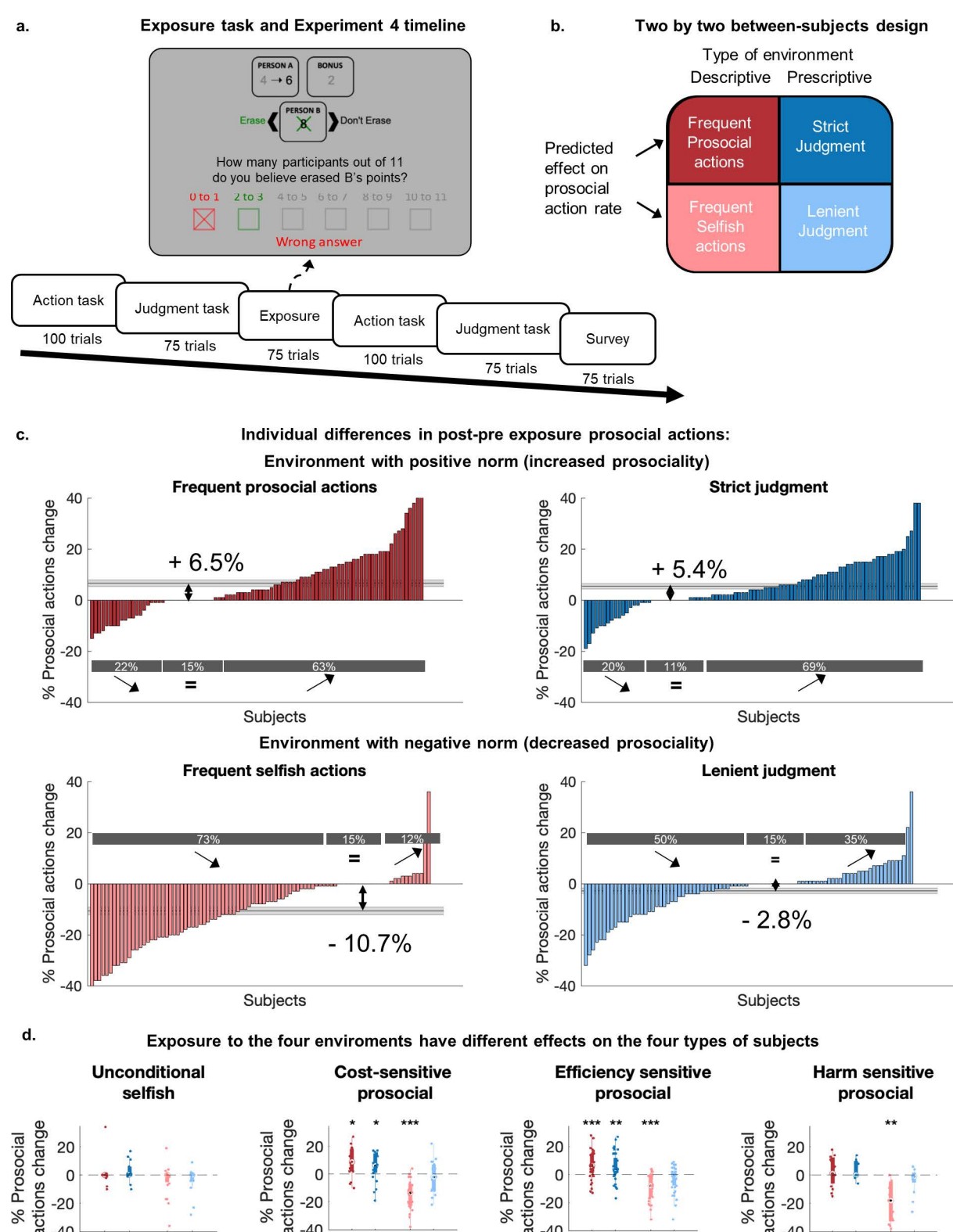

**Fig 4. Exposure to environments with different norms changes participants' behavior. (a)** Experiment 4 protocol. Timeline of the different tasks, and example trial for the frequent prosocial action environment. Participants guessed the behavior of previous participants and received feedback on the accuracy of their answer. **(b)** Two-by-two between-participant design. Each participant faced one of four environments. We varied the type of norm that

was manipulated (descriptive versus prescriptive norms) and the direction of the change (increased or decreased prosociality). **(c)** Individual differences in responses to the four environments. The differences in prosocial actions post- vs pre-exposure are displayed for each participant. The shaded grey line represents the mean and standard error across participants. The grey boxes show the percentage of participants who showed an increase, decrease, or no change in their prosocial action. **(d)** Prosocial action rate change post vs pre-exposure for the 4 different types of participants and 4 environments. Participants of Experiment 4 were clustered in the four groups previously described (using the clusters defined during Experiments 1 and 3), based on the pre-exposure parameters of the action model. The difference in prosocial action rate before and after the exposure is displayed for each cluster. The colors represent the 4 different environments. The stars represent the Bonferroni corrected significance of two-tailed t-tests (*: $p < 0.0125$, **: $p < 0.0025$, ***: $p < 0.00025$).

the previous participants was). To manipulate the norms, we provided feedback at the end of each trial, showing participants the behavior of selected participants of Experiment 3 who either behaved the most/least prosocially or who provided the strictest/most lenient judgments (S9 Fig, see S1 Text for details). Each participant was exposed to one of four environments: frequent prosocial actions (A+), frequent selfish actions (A-), strict judgment (J+), or lenient judgment (J-). To increase the credibility of our manipulations, we provided feedback based on the true behavior of participants of Experiment 3, rather than artificially constructing extreme behaviors (see S9c–S9e Fig and S1 Text for details about the structure and variability of the different environments).

Participants' pre-exposure actions and judgments largely replicated the results of Experiments 1 and 3 (S10 Fig). In this larger sample, the correlation between participants' behavior in the action and judgment task was again small but now statistically significant (S10 Fig, Pearson's correlation $r = -0.36$, $p < 0.001$). However, as in the previous experiment, a two-sample Kolmogorov-Smirnov test confirmed a significant difference in the distributions of the actions and judgments ($D = 0.86$, $p < 0.001$). Moreover, when comparing the correlations measured in Experiment 3 and 4 using a Fishers' transformation of the correlation coefficient, we found no significant difference between the two correlations (z-test: $z = 1.17$, $p = 0.084$). Thus, actions and judgments were only weakly related, in a similar fashion for both experiments.

Importantly, the participants' pre-exposure behavior was similar across the four treatment groups (S10 Table and S11 Fig). All four social environments significantly changed prosocial actions and judgments (Figs 4c, 4d and S12, Mixed-effects regressions of actions and judgments pre- vs post-exposure: S11 and S12 Tables, $p < 0.01$ for all environments). Exposure to frequent prosocial actions and strict judgments increased prosocial action rates to a comparable extent (A+: +6.5%; J+: +5.4%, Fig 4c and 4d, S11a Table interaction of norm type and experiment phase non-significant for positive environments: $p = 0.32$), showing that observing both prosocial behavior and prescriptions of prosocial actions promoted prosociality. In contrast, prosocial action rates decreased strongly after exposure to frequent selfish actions (A-: -10.7%) but to a much lower extent after exposure to lenient judgments (J-: -2.8%, S11d Table: Interaction of environment type and experiment phase for negative environments: $p < 0.001$). Thus, observing selfish actions had a large impact on the erosion of prosociality, while observing lenient judgments seemed to have more limited effects.

## Different types of individuals adapt differently to environments with different norms

Behavioral changes associated with the four normative environments differed strongly between participants (Fig 4d). These individual differences were neither explained by the pre-exposure behavior of the different participants (S13a Fig, correlation between prosocial action rate pre-exposure and prosocial action rate change p (Bonferroni corrected) > 0.4 for all groups) nor by the pre-exposure values of the individual motives (S13c–S13e Fig, correlation between pre-exposure parameters bias, $\gamma$ and $\mu$: p (Bonferroni corrected) > 0.07 for all groups and parameters). We therefore examined whether the four previously-identified types of participants responded differently to the four environments.

We classified participants of Experiment 4 using the clusters previously trained (See S7d and S7f Fig; replicating the clustering using Experiment 4 resulted in similar clusters: S14 Fig). All four types of participants understood the changes in norms in the new environments and adapted their judgments following exposure to frequent prosocial actions, frequent

selfish actions, and strict judgments (S15 Fig and S13a Table, ANOVA of the effects of the different environment types, directions and clusters on judgments changes p < 0.001 for environments types and directions, no significant effects of clusters or interactions between clusters and environments p > 0.4). However, exposure to lenient judgments did not significantly affect the judgments of participants of any cluster, showing that this environment failed to significantly shift the perception of the prescriptive norms for any type of participant.

Despite a homogeneous understanding of the changing social contexts, not all participants adapted their behavior. An analysis of variance showed significant effects of the interactions between the types of environments and types of participants (S13b Table, ANOVA: Interactions of environment types and clusters p = 0.029 and directions and clusters p = 0.007). *Unconditional selfish* did not adapt their actions following exposures to any environment (Fig 4d, post-hoc two-tailed t-tests: p > 0.15 for all environments). *Harm-sensitive participants* only adapted after observing frequent selfish actions: Their prosocial action rate decreased strongly (two-tailed t-test: p < 0.001), showing the strongest adaptation of all groups (- 19.1% against - 9.3% on average for other types). However, this group did not adapt their actions after exposure to other environments (two-tailed t-tests: all p > 0.13). Note that the lack of adaptation to some contexts may reflect ceiling or floor effects due to extreme pre-exposure actions (no adaptation in positive environments for *Harm-sensitive,* or in negative environments for *Selfish* participants). The remaining two types of moderately prosocial participants (*Cost-* and *Efficiency-sensitive*) changed their behavior in response to all environments (two-tailed t-tests: p < 0.015) except for lenient judgments (two-tailed t-tests: p > 0.015). Thus, *Harm-sensitive* participants appear particularly prone to prosociality erosion when they observe selfish others, whereas *Cost-* and *Efficiency-sensitive* participants appear to adapt their behavior in response to more diverse types of changes in their environment.

Further analysis of the changes in motives of the different individuals revealed that exposure to different environments did not affect the same decision mechanisms for all contexts and participant types (see S1 Text, S16 and S17 Figs, and S14 and S15 Tables), highlighting the importance of individual differences in these mechanisms.

Participants were additionally clustered based on their post-exposure motives. A four-by-four analysis of the pre- vs post-manipulation profiles revealed that most individual kept the same motivational type (73% of participants overall, 67% in the A+, 63% in the A-, 80% in the J+ and 81% J- environments), showing that motivational profiles are generally stable and that most changes in behavior reflect altered strength of individual strategies rather than shifts between strategies. However, some participants adopted different strategies. Most of these changes were heterogeneous and did not reflect systematic switches between specific strategies (see S18 Fig for a detailed description of these changes). One exception to this were *Harm-sensitive* participants, who shifted strategies more than other types: They frequently shifted to *Efficiency-* (33% of participants in the A+, 15% in the A-, and 38% in the J- environments) or *Cost-sensitive* (40% of participants in the A+ environment) strategies, showing that they adopted a less prosocial, more situation-dependent strategy when learning about the behavior of others.

## Discussion

Prosocial behaviors are crucial pillars of our societies and need to be promoted to address current challenges, including combating social inequalities, violent conflicts, or climate change. Key to the design of interventions that promote prosocial behavior is a solid understanding of the motives behind individuals' prosocial actions. In this study, we focus on the link between prosocial norms and prosocial decisions and propose a computational framework that can characterize the motives underlying individuals' perception of social norms and prosocial behaviors, with the aim of better understanding individual differences and diverse responses to contexts with changing social norms. Using computational modelling of behavior in tasks in which participants had to choose between selfish and prosocial actions or judge their appropriateness, we showed that different motives drive the perception of prosocial norms versus prosocial actions, and that understanding prescriptive norms alone does not guarantee prosocial behavior. Focusing on individual differences in prosocial motivations, a clustering approach allowed us to determine four different types of individuals who are guided by different motives and respond differently to contexts with changing social norms.

Norms are, by definition, widely considered standards of appropriate behavior. However, it has remained unclear how individuals' perception of norms translates into their actions [67]. While some studies have found that individuals' perceptions of prescriptive norms reliably correlate with group behavior [29], others suggest that prescriptions alone do not guarantee prosocial actions, and that participants comply with a norm only if they believe others also do so [22]. These studies, however, have focused on group-level behavior and have not studied whether an individual's perception of a norm could explain their actions. Focusing on individual behavior, our study shows that even though most participants agreed on what constitutes a prescriptive norm, prosocial behavior was not universal. In particular, although group judgments were strongly correlated with group actions as described previously [29], participants' own judgments were only weakly correlated with their own prosocial choices. Moreover, distinct motives influenced judgments versus actions, in line with research suggesting a distinction between personal and social norms [68]. Our results suggest that reporting on appropriate behaviors versus making a personal choice involves different cognitive and motivational processes. This discrepancy between understanding prescriptive norms and acting on them appears to mirror society's struggles to address issues such as climate change, where popularizing beliefs against meat consumption or plane travel does not always translate into the corresponding behavioral changes [69,70].

Our findings align with research on the neuroscience of norm compliance, which has highlighted fundamental differences between cognition (supporting the formation of beliefs and representations of decision situations) and valuation (allowing individuals to choose an action based on these beliefs and their individual preferences) [71], and they provide crucial evidence on how these differences are driven different underlying motives. The dissociation between individuals' judgments (relying on beliefs) and actions (relying on both beliefs and preferences) we report here may reflect that partially separated neural mechanisms support these functions. Brain activation and stimulation studies have indeed identified different brain regions involved in judgments versus normative decisions. The left dorsolateral prefrontal context (dlPFC) appears to be key for norm compliance but not for judgments, as stimulation of this region affects the participants' actions but not their perception of fairness [31,32]. By contrast, judging norm-violations appears to recruit other regions, such as the temporoparietal junction (TPJ) [72] and dorsomedial prefrontal cortex (dmPFC) [73], which are associated with the representation of social information [74]. Extending our framework to neuroscience experiments would allow us to directly test whether different (or only partially overlapping) sets of brain regions are recruited for actions versus judgments in the same situations, and to capture individual differences in these neural processes.

Our results additionally contribute to the debates on the nature of social preferences, and in particular the importance of contexts [50], outcome-based preferences [51,52], and unconditional moral preferences [75]. By using a combination of varying outcome ranges and computational models, we could capture the effects of different intrinsic motives within the same framework. First, we showed that the contextual framing we used had only a minor influence on prosocial decisions, compared to what had been previously reported [58–60]. While judgments were influenced by the framing of actions, actions mainly depended on choice outcomes. This refines theoretical proposals suggesting that coordination on normative judgments relies on categorical distinctions between types of actions, rather than on continuous considerations of outcomes [50]. Second, we found that models including both *outcome-based* and *baseline* preferences captured participants' behavior best, suggesting that both types of motives play a role in prosocial actions, but to different degrees in different participants. Similarly, in line with debates on the importance of efficiency in social decisions [52,56], we found considerable individual variability in participants' preference for efficiency. This may explain the variability in findings regarding the importance of this motive across studies [76,77]. Together, our results suggest that debates on the nature of social preferences that focus on group-level behavior alone may miss important aspects of human behavior linked to individual variability. The approach we propose here – developing comparable tasks and computational models of decisions across situations – is key for identifying such variability, as shown in recent studies of individual differences in motives underlying moral strategies [38,39,78] or distributive justice [33].

Social influences and desire for conformity play an important role in guiding behavior [79,80]. Studying these influences can help to explain the dynamics of prosocial behaviors – namely how they can be encouraged, or how they may erode – and could guide the design of behavioral interventions [41]. Although different types of normative messages have been successfully used to increase prosocial actions across contexts such as environmental conservation [20] or charitable giving [63], it has remained unclear whether information about the actions or judgments of others has a more reliable influence on prosocial behavior. Some studies comparing the effects of the two types of messages reported a higher influence of descriptive information on behavior [81], especially when the two messages are in conflict [21,22]. Others showed that the effects of the different messages may depend on individuals' traits such as cognitive elaboration [28]. Here we found that observing both the prosocial actions or strict judgments of others changed participants' prosocial decisions, confirming the strong effects of observed social norms [6,27,30,44] and conformity [80,81] on behavior. Contrary to previous literature [22], we did not find a marked difference in how strongly observing actions versus judgments increased prosocial actions.

The erosion of prosociality may also be strongly influenced by conformity mechanisms [82,83]. Observing peers behaving selfishly has been shown to increase crime [84], disorder [85] and dishonesty [86]. To our knowledge, however, no studies have directly compared the effects of prescriptive versus descriptive information on such changes. Our results showed considerable differences in the effects of observing actions versus judgments: Seeing others openly behaving selfishly appeared to encourage people to choose selfish actions, whereas hearing others approve of selfish actions only had a mild effect. In line with studies of peer influences on pro-social vs anti-social behavior [84], or the erosion of honesty [87] and cooperation [88], our findings emphasize the fragility of prosociality and highlight the necessity to understand the dynamics through which it can erode. Observing others behave poorly may lead individuals to feel justified or entitled to adopt selfish behaviors and stop them from exerting self-control. The same effect may be a lot weaker when we only hear others talk, without "putting their money where their mouth is". This may have consequences for the preservation of prosociality: To maintain prosocial norms, it may be less critical to prevent discussions about them, but essential to prevent openly selfish actions to avoid potential snowball effects.

While the global effects of changing environments on prosocial actions are noteworthy, not everyone reacts to these environments in the same way [89]. Using a systematic computational characterization of individual motivations, we found that individuals of different motivational profiles also differ in their adaptation to changes. This highlights that there may not be a one-size-fits-all solution for promoting prosociality. Systematic attempts to design behavioral interventions should consider the heterogeneity of profiles in the population. For example, in line with previous studies of the effects of social preferences on public good game contributions [90] or cooperation [91] changes, we showed that the most prosocial individuals (*Harm-sensitive*) were highly sensitive to selfish environments but did not adapt to prosocial environments. Such considerations are important for the design of behavioral interventions, in which decision-makers should avoid accidentally worsening the behavior of the most prosocial individuals [27]. Conversely, the most selfish individuals (*Unconditionally Selfish*) did not significantly change their actions following exposure to different environments, despite understanding the changing normative contexts. Extrinsic incentives may be necessary to push these individuals towards prosocial decisions [91], as they may comply only when it benefits them – as suggested by their high Machiavellianism [92] self-reported trait. Here we focused on the private motives underlying prosocial actions, but external influences such as reputation concerns [44] or punishment [9,42] may also be crucial for ensuring a high global level of prosociality. Future experiments should investigate the impact of reciprocity [93], second- [94] or third-party [43] punishment on various motives for prosocial actions, with a focus on examining the interplay between intrinsic and extrinsic motives and their individual variability. Understanding the responses to such external influences – and particularly those of *Unconditionally Selfish* individuals – would be essential for a comprehensive understanding of the factors that shape prosocial behavior. Computational characterization of population heterogeneity, with approaches such as the one we propose here, may be key for achieving this goal.

Our computational approach identified four types of participants in our specific experimental population: 41% of participants were *Efficiency-sensitive Prosocial*, 24% *Cost-sensitive Prosocial,* 20% *Unconditionally Selfish*, and 15% *Harm-sensitive Prosocial*. Taking this distribution into account, interventions can be tailored to specific target groups based on their composition. In our university student sample, the majority of participants belonged to the two mildly prosocial groups who adapt their behavior when being exposed to both actions and judgments. This may explain the high overall behavior change following exposures. Other target populations may have different compositions [95,96]. For instance, a recent set of studies characterizing types of other-regarding preferences in Swiss society identified three types of individuals: Altruistic, inequity-averse, and selfish [97]. Comparisons of the distributions of these three types in samples from the Swiss population versus Swiss students revealed that there were almost no inequity-averse students, whereas this type represented almost 50% of the general population [96]. This highlights the necessity of characterizing the composition of diverse target groups. It also remains to be shown whether the same types of individuals can be found in different societies. The computational approach we established here can, in principle, be applied to diverse populations and cultural contexts [98,99], thereby helping to identify different motivational types and normative behavior across different cultural settings.

## Materials and methods

### Ethics statement

The study was approved by the Human Subjects Committee of the Faculty of Economics, Business Administration and Information Technology at the University of Zurich (OEC IRB # 2019–026).

### Experimental tasks

**Time estimation task.** Participants first played a time estimation task to collect points, making decisions closer to real-life contexts in which people work to earn their payment. In each trial, participants estimated an amount of time and could collect 2 to 10 points, depending on their accuracy and a hidden maximum score (see S1 Text and S1a and S1b Fig for a detailed task description).

**Action task.** We designed an action task to quantify the different intrinsic motives underlying prosocial actions. In each trial, two participants (Player A and Player B) were paired randomly and anonymously. Players A were shown the points they and Player B earned during a time estimation trial and offered a bonus of 2, 4, or 6 points. Taking the bonus resulted in Player A earning their time estimation points, plus the bonus points, and Player B earning 0. Refusing the bonus resulted in both players earning their time estimation points. The decisions were framed as either Helping or Destroying, depending on the context. In the Helping frame, Player B's points were erased by a lottery and Player A could reverse this outcome by refusing the bonus, allowing B to recover their points. In the Destroying frame, the lottery did not erase Player B's points and Player A had to actively take these points away to get the bonus. Importantly, regardless of their decisions, Player A could never lose their originally earned payoff ($p_A$) and only made decisions regarding additional gains for themselves (bonus), versus loss for the other player ($p_B$). Two versions of this task varied in the way information was presented to Player A (see S1 Text). The results of these two versions were not significantly different (S1 Table, version $p > 0.1$) and therefore pooled for analysis.

Participants played as many combinations as possible of their own ($p_A$) and Player B's ($p_B$) outcomes, and bonus (b), given their performance on the time estimation task (Fig 1d). In Experiment 1, Player A faced the two contexts, in Experiments 3 and 4, participants only faced the Destroying context. At the end of the experiments, some trials were randomly selected and were paid to the participants (1 point = 1 CHF).

**Judgment task.** To measure normative expectations, we used an adaptation of the judgment task proposed by Krupka and Weber [29]. Participants evaluated the appropriateness of fictive Player A's decisions in the action task. They viewed all possible situations and decisions of Players A (prosocial and selfish actions) and rated the appropriateness of these

decisions on a 6-item Likert scale, ranging from *very socially inappropriate* to *very socially appropriate* (Figs 1c, S1e and S1f). Participants were instructed to respond as truthfully as possible, based on their opinions of what the majority would consider socially appropriate or inappropriate behavior. They were incentivized to match the responses of other participants in the room: At the end of the experiments, some trials were randomly selected, and each participant earned 5 CHF per trial if the answer they selected matched the most frequently selected answer among the other participants for this situation.

**Exposure to environments with different norms.** To evaluate participants' responses to contexts with changing social norms, we designed a task in which participants had to guess the behavior of previous participants. We manipulated two types of norms: Prescriptive norms for which participants had to guess the appropriateness ratings given by previous participants of the judgment task, and descriptive norms for which participants had to guess how many previous participants chose the selfish action in the action task. They viewed all possible situations (Fig 1d) and were given feedback on the behavior of 11 selected participants of Experiment 3. We manipulated the normative context by selecting participants with the highest (or lowest) frequency of selfish action or the strictest (or most lenient) appropriateness judgments. This allowed us to create four different contexts: Frequent prosocial actions, frequent selfish actions, strict judgment, and lenient judgment. The details of the correct answers for the different trials for each type of exposure are shown in S9c Fig. At the end of the experiment, one trial was selected, and participants were paid 5 CHF if they had guessed correctly on this trial.

**Demographic and sub-clinical questionnaires.** At the end of the experiment, participants completed a demographic questionnaire. Participants of Experiments 3 and 4 also responded to several personality and subclinical questionnaires (S9 Table).

## Participants

All participants were recruited from the UZH Department of Economics participant pool and provided written informed consent. Each participant participated in only one of our four experiments (see S16 Table for a summary of the four experiments). There was no deception.

We recruited a total of 575 participants. All participants played the time-estimation task, followed by one or several of the other tasks. In Experiment 1, 75 participants played up to 300 trials of the judgment task. In Experiment 2, 70 participants played up to 300 trials of the action task. In Experiment 3, 72 participants played 150 trials of each task (tasks played in a counterbalanced order). In Experiment 4, 358 participants played 75 trials of the judgment task, 100 trials of the action task (in a counterbalanced order) followed by 75 exposure trials, and 100 and 75 trials of the judgment and action tasks again. For all experiments, we also recruited participants as Player B who participated in the time estimation game and were paid according to the decisions of Players A. The details of the different session and payment schemes for each experiment can be found in S1 Text and S16 Table.

## Statistical analysis

All the statistical analyses were conducted using R (R Core Team 2017). Mixed-effects models included random intercepts for each participant, as well as random slopes for all regressors corresponding to within-participant manipulations (except for ordinal regressions, for which some random slopes hindered model convergence, see details below).

We analyzed the decisions of Players A in the action task (*Prosocial action*) using binomial Generalized Linear Mixed Effect Models (*glmer* with probit link from *lme4* package).

We analyzed the ratings of the judgement task using mixed effect ordinal regressions (*clmm* from *ordinal* package). Due to the complexity of ordinal data fit (the thresholds between each rating must be estimated), we could not include all

random slopes in these models. We systematically included a random intercept for each participant, and a random slope for the two contexts. Models including random slopes for the different payoff and bonus values did not converge (See S1 and S3 Tables for details).

We compared the models described above with models including demographic variables using the ANOVA function in R. Including demographic variables such as age, gender, monthly spending money, political orientation, and religion as regressors slightly improved our model fits but did not change the coefficients of the other independent variable (See S1 and S2 Tables).

## Computational modeling of social decisions

**Modeling choices and judgments.** We analyzed actions and judgments using computational utility models including different motives. We extended a classical model of social preferences proposed by Charness and Rabin (CR) [52] to quantify the sensitivity of each participant to different motives.

The most general model proposed by Charness and Rabin to includes two parameters: $\gamma$ reflecting *outcome-based* preferences for prosocial versus selfish outcomes, and $\mu$ reflecting consideration for the worst-off player (*harm-aversion*) versus *efficiency*. We extended this model to measure *context-based* and unconditional preferences, using two parameters: A bonus discount factor ($\delta$) reflecting specific context influences and a bias term reflecting *baseline* preferences (See S5 Table for utility computation details).

We compared this full model systematically to models that did not include a bias, or a *context* factor, as well as models that relied on different motives. The models we included were based on the Fehr-Schmidt [51] model, reflecting an aversion for an unequal repartition of money between the two players, a harm-aversion [57] inspired model balancing the bonus vs the payoff of Player B, and the model proposed by Krupka and Weber (KW) that uses the appropriateness ratings collected during the judgement task to predict the behavior of Players A facing the same situation (see S1 Text and S5 Table). Note that the average appropriateness ratings vary across choice situations, capturing differences between contexts that may influence Player A's decision. We therefore did not enrich the KW model with a bonus discount factor, as this parameter would be colinear and could not capture additional unique influences of contexts. A model enriching the Fehr-Schmidt model with efficiency has been used in previous research but was later shown to be equivalent to simpler models and not identifiable [100]. We therefore did not include this model variant in our analysis.

To evaluate whether the same decision processes are employed in deciding what constitutes an appropriate behavior versus acting upon it, we used the same model of utility differences between the prosocial and selfish actions of Player A to account for both the actions of Player A and the judgments emitted by an observer of such actions (from very socially inappropriate to very socially appropriate). The only difference was the transfer function translating the motives to actions (sigmoid (SoftMax) computing the probability of choosing the prosocial action) versus judgements (ordered-probit function [101] computing the probability of choosing each of the 6 ratings; see S1 Text).

**Model fitting and comparison.** We used a hierarchical Bayesian framework to fit the different models to the participants' behavior and estimated participant-specific parameter values as the mode of the participant-level parameter posterior distribution. We used a method approximating leave one out cross validation [66] for model comparison (See S1 Text for hierarchical fitting and model comparison details). The models were implemented in R and fitted using JAGS [102].

**Parameter recovery.** We used the fitted parameters of the data from Experiments 1 and 2 to simulate choices using the winning model (CR $\delta 1$ bias) and fitted the model to these simulated choices. All parameters of the action and judgment models recovered well (S6b Fig, correlations between simulated and recovered parameters $r > 0.44$, $p < 0.001$ for all parameters)

**Clustering.** We used the fitted model parameters of each participant to cluster them in different categories. We used the estimated parameters (*bias*, $\gamma$, $\mu$) values of the winning model, and the inverse temperature of the SoftMax function

to define different clusters using a K-means algorithm. We used a data-driven approach to optimize the number of clusters (S1 Text).

## Supporting information

**S1 Text. Supplementary information.**
(DOCX)

**S1 Fig. Design of the different versions of the tasks.** Supplement to Fig 1. (**a**) Time-estimation task: In the first part of Experiments 2–4, both players A and B collected points on 300 trials of this task. They were asked to estimate a specified amount of time by pressing the space bar on their keyboard at the accurate amount of time after a cue (central cross) turned red. (**b**) Time estimation task: Results of Experiment 2 and payoff scheme. In the top plot, each cross represents the estimation of a participant of Experiment 2, on a trial. The dashed red lines represent the boundaries outside of which the participants received the minimum number of points. The black line represents the perfect estimation. The closer the estimations were to this black line the more points participants collected. The number of points available on each trial varied across trials, the bottom left plot represents this points distribution across all trials. The participants collected a fraction of the number of points available, depending on their accuracy. The resulting distribution of points collected across participants and trials is displayed in the bottom right plot. (**c**) Version 1 of the action task (Experiment 2). Participants were reminded of the time estimation trial they faced and the points they collected. They were then shown the points player B collected. The lottery was computed on every trial and could erase the points of Player B. Player A was then offered some bonus points and decided whether to take or reject this bonus, with consequences for the outcome of Player B. (**d**) Version 2 of the action task (Experiment 2) as described in Fig 1, the trials were grouped in blocks of 10 trials between which the lottery was recomputed. Participants were not reminded of the time estimation trial that yielded their points. They decided to Erase/Not Erase (Destroying frame) or Help/Not Help (Helping frame), instead of deciding to Take/Not Take the bonus. The consequences of participants' actions remained the same as in Version 1. In Experiments 3 and 4, there was no lottery, and participants only faced Destroying trials, the rest of the task was displayed identically. (**e**) In-person version of the judgment task. Participants were shown situations in which Player A chose a prosocial or selfish action (Helping/Not Helping or Not destroying/Destroying). Participants answered a series of 3 questions per situation – one for each bonus value (2, 4, and 6). The actions of Player A were described as "take" or "not take" the bonus, as in Version 1 of the action task. (**f**). The online version of the judgment task is described in Fig 1. The bonus values were randomized across trials, participants replied to one question at a time, and the actions of Player A were framed as "Help/Not Help" and "Erase/Not Erase".
(TIF)

**S2 Fig. Appropriateness judgments of prosocial actions did not depend on payoff distributions or contexts.** Supplement to Figs 1 and 2, judgment data from Experiment 1. (**a**) Distributions of ratings for selfish and prosocial actions, across all participants and trials of Experiment 1, showing low individual differences. All judgments were rescaled to 0–1 interval: 0 corresponds to very socially inappropriate and 1 to very socially appropriate. (**b**) Average appropriateness judgment of prosocial actions for each participant. The blue line and interval represent the mean and its confidence interval across participants. (**c**) Effect of payoff distributions on normative judgments of prosocial actions and on its difference between contexts. The color scales represent the average appropriateness judgment of trials during which A chose the prosocial action (left), or the difference in these ratings between the two contexts (right). The black lines represent efficiency thresholds: Selfish actions are efficient for trials above the dotted lines, and prosocial actions are efficient for trials below the dashed line. Trials between those two lines have equal efficiency for selfish and prosocial actions ($pB = b$). (**d**) Absence of effects of selfish action efficiency (b-pB) and score of the worst-off player (min ($pA$, $pB$)) on mean appropriateness judgment of prosocial actions. These results show the low inter-individual and inter-trial variations of judgments

provided when Player A picked the prosocial action, showing that prosocial action is viewed as unanimously appropriate regardless of the specific situation. (**e, g**) Individual differences in judgments (rescaled to 0 (very socially inappropriate) - 1 (very socially appropriate)) for trials in which Player A picked the selfish action (e) or actions of Player A (g). Each bar represents the average rating of a participant of Experiment 1 (e) or the prosocial action rate of a participant of Experiment 2 (g); the different colors indicate which version of the experiment participants participated in (see S1 Text and S1 Fig). The blue line and interval represent the mean and its confidence interval across participants. (**f, h**). Effect of payoff distribution (points of Player B and bonus) on differences between contexts for judgments (f) or actions (h). The color scales represent the average behavior across participants of Experiment 1 (f) or 2 (h). The black lines represent efficiency thresholds: Selfish actions are efficient for trials above the dotted lines, and prosocial actions are efficient for trials below the dashed line. Trials between those two lines have equal efficiency for selfish and prosocial actions ($p_B = b$).
(TIF)

**S3 Fig. Replication of the main behavioral patterns of actions and judgments in Experiment 3.** Supplement to Fig 2. (**a**) Individual differences in judgments of trials in which Player A picked the selfish action. Each bar represents the average rating of a participant (rescaled to 0 (very socially inappropriate) to 1 (very socially appropriate)). (**b**) Individual differences in prosocial actions. Each bar represents the prosocial action rate of a participant in the experiment. (a-b) The colors indicate whether participants played the action (red) or judgment (blue) task first. The blue line and interval represent the mean and its confidence interval across participants. (**c, d**) Effects of prosocial action Efficiency (pB - b) and score of the worst-off player (min (pA, pB)) on the mean appropriateness judgments of selfish actions rescaled to 0–1 interval (c) and prosocial action rate (d) for the two different task orders: Action 1st in red and judgment 1st in blue. Each cross represents a participant, the lines are the average across participants and error bars represent the standard error of the mean. Participants who participated in the judgment task 1st have an increased prosocial action rate, and stricter judgments compared to participants who started with the action task (S1 and S2 Tables), nonetheless, the effects of the different task parameters are comparable in these two groups. (**e, f**) Effect of payoff distributions (points of Player B and bonus) on average judgments (e) and prosocial action rate (f). The color scales represent the average behavior across participants. The black lines represent efficiency thresholds: Selfish actions are efficient for trials above the dotted lines, and prosocial actions are efficient for trials below the dashed line. Trials between those two lines have equal efficiency for selfish and prosocial actions (pB=b). selfish action increases with stakes (bonus) and decreases with potential harm caused (pB). Judgments become more lenient (towards appropriate) as stakes (bonus) increase and stricter (towards inappropriate) as the harm caused increases (pB). (**g**) Relation between selfish action frequency and appropriateness judgments averaged across participants of Experiment 3, for the 75 different trials. The colors represent the task orders (action 1st in red and judgment 1st in blue). Each dot represents the results of one trial, averaged across all participants. The lines and coefficients show regressions between selfish actions and judgments. As shown in Fig 1, judgments averaged across participants are strongly correlated with the average selfish action rate. These results replicate the findings of Experiments 1 and 2 and show that participating in both tasks did not significantly alter participants' patterns of behavior. (**h**) Bivariate distribution of fitted parameters for participants of Experiment 3. The mean and standard deviations across participants are represented in plain (resp dotted) lines. The black dots represent participants and show low but significant correlations between the action and judgment parameters.
(TIF)

**S4 Fig. Extended Charness and Rabin model captures participants' behavior best.** Supplement to Fig 2. (**a**) Experiment 2: Measured and predicted behavior using simulations of the different models (using the estimated parameters of each participant): Charness and Rabin model with a bias and discount factor (CR $\delta$1 bias), Krupka and Weber model (KW), modeling actions using judgment data, and Krupka and Weber model including a bias (KW bias). The lines represent data simulated using the fitted model parameters of each participant. The crosses are the measured behavior. The

colors correspond to the two contexts: Not destroying in pink and Helping in purple. The KW model failed to capture the effect of the points of Player B on prosocial actions, whereas the CR model best explained the participants' choices. (**b**) Experiment 2: Difference between the behavior measured and simulated using the different models. The colors represent the difference in proportion of trials where Player A chose the prosocial actions. Blue colors represent a model under-estimation of the prosocial decisions, and red colors a model overestimation of prosocial choices. Black lines represent efficiency thresholds: Selfish actions are efficient for trials above the dotted lines, and prosocial actions are efficient for trials below the dashed line. The Krupka and Weber models do not capture the effect of efficiency on the choices of Player A, while the Charness and Rabin model shows no systematic deviation from the measured behavior. (**c**) Experiment 3: Measured and predicted behavior using simulations of the different models: Charness and Rabin including a bias term (CR bias in green), Krupka and Weber using the average judgment across all participants (KW group in orange), or the judgment of the participant making the actions (KW individual in purple). The lines represent data simulated using the fitted model parameters of each participant and the crosses represent the measured behavior, showing again that the KW model failed to capture major trends in behavior.
(TIF)

**S5 Fig. Model simulations: Extended Charness and Rabin model flexibly captures different behavioral patterns.** Simulations of the best model (CR $\delta 1$ bias) illustrate the effects of parameter variations on behavior. The color scales represent the predicted proportion of prosocial actions, the black lines represent efficiency thresholds: Selfish actions are efficient for trials above the dotted lines, and prosocial actions are efficient for trials below the dashed line. Trials between those two lines have equal efficiency for selfish and prosocial actions ($p_B = b$). These simulations show that the different parameters model different aspects of behavior reflected in the influence of the payoff distributions on partici-pants' choices, which can be observed in the experimental population (see Fig 3d for the true behavior of different types of participants).
(TIF)

**S6 Fig. Independence and recovery of model parameters.** Supplement to Fig 2. (**a**) Relations between the four parameters of the winning model (CR $\delta 1$ bias): *Context-based* preferences ($\delta 1$), *outcome-based* preferences ($\gamma$), *specific goals* ($\mu$), and *baseline* preferences (bias). Each dot represents the parameter of one participant. The black lines and coefficients represent regressions between the different parameters. None of the parameters are significantly related, showing that the model captures distinct aspects of behavior. (**b-c**). Parameter recovery action model (b) and judgment model (c). Generating parameters (empirical values fitted to the data of Experiments 2 (b) and 3 (c)) are used to simu-late choices using the CR $\delta 1$ bias model. These choices are used to recover the generating parameters (using JAGS for hierarchical model fitting, as in the main analysis). Each dot plots the recovered parameter from the simulated behavior against the generating parameter, and the black lines represent the regression of the recovered on the generating param-eters. All parameters recovered well, validating our modeling approach.
(TIF)

**S7 Fig. Clustering results of Experiment 2 and Experiment 4 reproduce the results obtained in Experiment 3 and show the stability of behavioral types across different participant samples.** Supplement to Fig 3. (**a** and **d**) Parame-ter combinations of each participant of Experiment 2 (a) and Experiment 4 (d) and centroid of the 4 clusters resulting from the optimization procedure and training on Experiment 1 and 3 data. Four different types of participants are represented by different markers and colors. (**b** and **e**) Number of participants in each cluster, for Experiment 2 (b) and Experiment 4 (e). (**c** and **f**) Parameter distributions of the 4 clusters for participants of Experiment 2 (c) and Experiment 4 (f), effects of bonus and points of player B on prosocial actions averaged across participants in each of the four clusters. The colors represent the prosocial action rates. The black lines represent efficiency thresholds: Selfish actions are efficient for trials above the dotted lines, and prosocial actions are efficient for trials below the dashed line. Trials between those two lines

have equal efficiency for selfish and prosocial actions ($p_B = b$). Participants of the different clusters have different patterns of prosocial actions, reflecting the importance of parameters for their actions. The patterns of prosocial actions are similar across all experiments for the different clusters, confirming the relevance and stability of the clustering procedure. (TIF)

**S8 Fig. Demographic and personality traits and their correlation with normative behaviors.** Supplement to the results section "Four types of individuals with distinct patterns of motives" (**a**) Statistically significant (using a Bonferroni correction $P < 0.0083$) Pearson's correlations between self-reports and average behaviors in the action and judgment tasks. Each dot represents a participant of Experiments 3 and 4. The demographic and personality traits are measured using the questionnaires reported in S9 Table. No trait was significantly related to average judgments, while action rates correlated with perspective taking and empathic concern subscales of the IRI questionnaire, and were anticorrelated with primary psychopathy, Machiavellianism, and right-wing orientation. (**b-f**) Statistically significant (using a Bonferroni correction $P < 0.0125$) Pearson's correlations between self-reports and fitted parameters of the action and judgment models. Each dot represents a participant of Experiments 3 and 4. (b) Action model, *baseline* preferences (bias). (c) Judgment model, *baseline* preferences (bias). (d) Judgment model, *specific goals* ($\mu$). (e) Action model, *outcome-based* preferences ($\gamma$). (f) Judgment model, *outcome-based* preferences ($\gamma$). (**g**) Differences in task performance and self-reports of participants of the different clusters. Each bar is the mean score across participants of a cluster. The significance level corresponds to $P$-values $< 0.125$ (Bonferroni corrected) of post hoc two-tailed t-tests comparing the score of one versus all 3 other groups when the clusters have significant effects on the score (tested with ANOVA, $P < 0.05$). These results show that some traits such as psychopathy, Machiavellianism, or empathic concern are reflected in participants' normative behaviors and related to specific actions and judgments concerns. The type of participant *Unconditional Selfish* has, in particular, high psychopathy and Machiavellianism and low empathy or altruistic scores. (TIF)

**S9 Fig. Details of the four different normative environments presented to participants of Experiment 4.** Supplement to Fig 4. (**a**) Exposure screen for the descriptive norms (observing actions). Participants must guess how many out of 11 previous participants decided to take the bonus and erase the points of Player B in the situation described. (**b**) Exposure screen for the prescriptive norms (observing judgments). Participants must guess the mode appropriateness rating of 11 previous participants judging the actions of a fictive Player A in the situation described. (**c**) Correct answers for the frequent prosocial action and frequent selfish action environments (descriptive norms). The color gradient shows the correct responses, corresponding to how many participants picked the selfish action in the different situations, as a function of the bonus and points of Player B. The correct responses were computed using the behavior of the most (resp. least) prosocial participants of Experiment 3 and rescaled to 1–6. (**d**) Correct answers for strict and lenient judgment environments (prescriptive norms). The color gradient shows the correct responses, corresponding to the mode appropriateness ratings of 11 previous participants in the different situations, as a function of the bonus and points of Player B. The correct response was computed using the mode behavior of the participants of Experiment 3 judging selfish action as the most (resp. least) inappropriate on average. (**e**) Percentage difference in the average actions or judgments between the four selected groups and the average of all participants of Experiment 3, for the different values of the bonus and points of Player B. The colors represent the difference between the group and overall mode behavior. (**f**) Learning performance. Percentage of correct answers during the exposure phase for each environment. Each bar represents a participant. The colors represent the task orders (action 1st in blue and judgment 1st in red). The grey line and shaded areas are the mean and standard error across participants. This figure shows the differences in the feedback provided in the four environments in the exposure phase. See S1 Text for a detailed discussion of these differences and their potential impact on behavior. (TIF)

**S10 Fig. Participants' behavior in Experiment 4 replicates Experiment 1–3.** Supplement to Fig 4. (**a and b**) Effects of prosocial action efficiency ($p_B$-b) and score of the worst-off player (min ($p_A$, $p_B$)) on the proportion of prosocial action trials (a) and mean appropriateness judgment of selfish action trials rescaled to 0–1 interval (b). For the two different task orders: Action 1st in red and judgment 1st in blue. Each cross represents a participant, the lines are the average across participants and error bars represent the standard error of the mean. (**c**) Correlation between selfish action rates and appropriateness judgments averaged across participants of Experiment 4, for the 75 different trials. The colors represent the task order: Action 1st in red and judgment 1st in blue. Each dot represents the results of one trial, averaged across all participants. The lines show regressions between selfish actions and judgments. Judgments averaged across participants are strongly correlated with the average selfish action rate, as reported in Experiments 1,2 and 3. (**d**) Bivariate distribution of the prosocial action rates (left), and mean appropriateness judgments (right), for each participant of Experiment 4. The mean and standard deviation across participants are represented in plain (resp. dotted) lines and show wide individual differences in prosocial action, and relative consensus of appropriateness judgments, as noted in Experiment 3. The black crosses represent each participant and show a small correlation between the average actions and judgments of the different participants. Although significant, this correlation is much smaller than the correlation at the group level (c), showing that average judgments have low predictive power on individuals' prosocial actions. (**e**) Relation between prosocial action and judgment changes post- vs pre-exposure, showing a small effect. Participants who adapted their judgments the most after the exposure also had a higher change in their prosocial action rate.
(TIF)

**S11 Fig. Actions and judgments of the four different groups are comparable before the exposure phase.** (**a and c**) Effect of payoff distributions (points of Player B and bonus) on prosocial action frequency (a) and average judgments (c) for the 4 groups, before the exposure phase. The color scales represent the average behavior across participants. (**b**) Individual differences in prosocial action for the 4 groups, before the exposure phase. Each bar represents the prosocial action rate of a participant before the exposure phase, the colors indicate whether participants played the action (red) or judgment (blue) task first. The blue line and interval represent the mean and its confidence interval across participants. (**d**) Individual differences in normative judgments for the 4 groups, before the exposure phase. Each bar represents the average rating of a participant (rescaled to 0 (very socially inappropriate) to 1 (very socially appropriate)). These results show that participants' behavior in the four different experimental groups is comparable before the exposure phase, allowing us to compare the effects of the different types of exposure.
(TIF)

**S12 Fig. Changes in judgments post- versus pre-exposure show that participants of Experiment 4 successfully learned the changes in normative environments.** Supplement to Fig 4. Individual differences in responses to the four environments. The differences in judgments post- vs pre-exposure are displayed for each participant. The shaded grey line represents the mean and standard error across participants. The grey boxes show how many participants showed an increase, decrease, or no change in their judgments. Except for the lenient judgment exposure, most participants adapted their judgments in the predicted direction.
(TIF)

**S13 Fig. Relations between action changes post-exposure and pre-exposure behaviors and motives for participants of Experiment 4.** Supplement to Fig 4. (**a** and **b**): Relations between pre-exposure behavior and action change post vs pre-exposure. Each dot represents a participant and the colors represent the task orders. (**a**) Action: Absence of effects of the pre-exposure prosocial action rates on action changes post vs pre-exposure. (**b**) Judgments: Participants exposed to prescriptive norms (guessing the judgments of previous participants) had a significant correlation between their baseline judgments and prosocial action changes. For participants seeing strict judgments: The more lenient their baseline judgments were, the more their prosocial action increased after the exposure. Participants who were far off from

the strict definition of the norm changed the most. In the lenient judgment environment, the stricter participants were in their baseline judgment (the further off from the lenient normative environment), the more their prosocial action decreased. Participants who had a perception of the prescriptive norm far from the one displayed during the exposure phase changed their behavior the most. These effects are however small, showing that the changes in prosocial action are mostly independent of the aggregate prosocial action or judgments pre-exposure. (**c-e**) Relations between the pre-exposure parameters of the CR bias model and action changes. Bonferroni-corrected correlations show no significant relation (all $p > 0.004$) between individual parameters and change in behavior. Each dot represents a participant, colors represent the order of the two tasks and the lines regressions between the two variables.
(TIF)

**S14 Fig. Clustering control:Using data from Experiment 4 exclusively replicates similar clusters.** Supplement to Fig 3. The clustering algorithm training and optimization of the number of clusters was repeated using Experiment 4 only. (**a**) Parameter distributions of participants of Experiment 4 clustered using the clusters trained on data from Experiments 1 and 3. (**b**) Parameter values in the 5 new clusters found when repeating the clustering procedure using data from Experiment 4 only. Some clusters were equivalent to the previously defined clusters, while the *Efficiency-sensitive* cluster was separated into two clusters, that differed in their bias values. (**c**) Prosocial choices of the 5 newly defined clusters, with varying points of B and bonus. Most clusters had a similar pattern as the ones described in Fig 4. The two new sub-clusters containing the previous *Efficiency-sensitive* participants showed similar responses to variations of the bonus values and points of Player B, as well as a similar overall prosocial action rate. (**d**) Number of participants in each of the 5 newly defined clusters. (**e**) Bivariate distribution of participants of Experiment 4, across clusters defined in Experiments 1 and 3 (old clusters) and clusters defined in Experiment 4 (new clusters). Most participants that belonged to an « old cluster » are grouped in the same « new cluster », showing the robustness of the clustering procedure: Similar clusters are produced when training the algorithm using different sets of participants. As shown in (b) participants belonging to the *Efficiency-sensitive* cluster were separated into two distinct clusters (1 and 2) using this new data set. These results show the stability of our clustering approach, as replicating the procedure with a new data set generated similar clusters.
(TIF)

**S15 Fig. Different types of participants have different responses to exposure to new normative environments: Judgment changes.** Supplement to Fig 4. (**a**) Post vs pre-appropriateness judgments of selfish actions for the 4 different types of environments. Participants of Experiment 4 were clustered into four groups, based on the baseline parameters of the action model. The difference in judgments before and after the exposure is displayed for each cluster. The colors represent the 4 different environments: A+ (Action+, frequent prosocial action) and J+ (Judgment+, strict judgment) are normative environments increasing prosocial actions, and A- (action -, frequent selfish action) and J- (Judgment -, lenient judgments) normative environments decreasing prosocial actions. Each dot represents a participant. The box overlay represents the median and 25% and 75% quartiles. Most individuals adapt their judgments after exposure. The changes in the four groups are comparable. (**b**) Judgment parameter changes following exposure. The CR model predicting judgments was modified to include parameter changes, that represent how much each parameter evolved after exposure. The colors represent the 4 different clusters, and each parameter change (*baseline* preferences (bias), *outcome-based* preferences ($\gamma$), and *specific goals* ($\mu$)) is displayed for each cluster and environment. Each dot represents a participant. The box overlay represents the median and 25% and 75% quartiles. The patterns and magnitude of parameter changes are similar for the different clusters within each normative environment, although some parameter changes are below significance.
(TIF)

**S16 Fig. Different types of participants have different responses following exposure to new normative environments: Actions changes.** Supplement to Fig 4. Action model (CR bias) parameter changes following exposures. The

CR bias model was modified to include a measure of parameter changes, that represent how much each parameter evolved after exposure. (**a**) Parameter changes across all participants, the colors represent the four environments and (**b**) parameter changes for each cluster, the colors represent the 4 different clusters. Each parameter change (*baseline* preferences (bias), *outcome-based* preferences (γ), and *specific goals* (μ)) is displayed for each cluster and environment. Each dot represents a participant. The box overlay represents the median and 25% and 75% quartiles. This figure shows that distinct types of participants show different patterns of change. See S15 Table for an Analysis of variance (ANOVA) of the effects of the types of participants (cluster), type of environment (descriptive versus prescriptive norms), and direction (positive versus negative changes) on the average changes in parameters. (TIF)

**S17 Fig. Pre-exposure parameters and parameter change post vs pre-exposure for the 4 environments.** Supplement to Fig 4. Each dot represents the parameters of one participant. The colors represent the different normative environments. For environments increasing prosocial action, the higher the baseline *outcome-based* preferences (high γ), the more this parameter increases after the exposure. Participants sensitive to prosocial outcomes are inclined to become more sensitive to prosocial outcomes. Similarly, the lower the baseline *outcome-based* preferences, the more selfish participants become when observing frequent selfish actions. Participants observing lenient judgment mostly increase their *outcome-based* preferences, the higher the parameter, the more this parameter increases. For most environments (except observing frequent prosocial action), participants that have a baseline behavioral goal of *efficiency* (low μ) increased their efficiency concern even further. Pre-manipulation *baseline* preferences (bias) only significantly correlated with a bias increase when participants observed strict judgments: The more prosocially biased they were, the more this bias increased. The general trend shows that the more extreme participants' preferences are, the more exposure to environments in agreement with these preferences reinforces them. (TIF)

**S18 Fig. Change in cluster post exposure: Confusion matrix.** Participants of experiment 4 were clustered again based on the result of a CR bias model fitted to their post-exposure decisions. The pre- and post- exposure clusters were compared. The colors and numbers represent the percentage of participants of each pre-exposure cluster being clustered in each of the 4 types post exposure, for the 4 different environments. Most participants remained in the same cluster, showing relative stability of the motivational profiles. However, there were some systematic changes across the minority of people who shifted their motivational profile in response to the change in environment. *Unconditional selfish* mostly remained *Unconditional selfish* regardless of the environment; only a minority of them shifted towards a *Cost-sensitive* or *Efficiency sensitive* profile when observing strict judgments. A third of *Cost-sensitive* participants shifted their motivations towards *efficiency* when witnessing frequent prosocial actions, but also lenient judgments, while a third of them became *Unconditional selfish* when witnessing frequent selfish actions. The *Efficiency-sensitive* type was mostly stable, but did shift to *Unconditional selfish* or *Harm-sensitive* types when witnessing frequent selfish actions, showing that the response of these participants to this environment could strongly vary. Finally, *Harm-sensitive* participants changed types the most often and shifted towards more mitigated strategies (*Cost-* or *Efficiency- sensitive*) when witnessing prosocial actions, or even to *Unconditional selfish* when witnessing frequent selfish actions. (TIF)

**S1 Table. Statistical analysis – Judgments Experiment 1 and 3 main effects.** Fixed effects coefficient estimates, standard errors, and p-values of the judgment regressions mixed-effects models using participants as random effects. The judgment data were analyzed using a Cumulative Link Mixed Model. The judgments were analyzed separately for trials in which the fictive Player A chose the prosocial (a) versus selfish action (b). Due to model convergence issues, only the effect of the context, and bonus (for selfish trials) were included as a random slope for each participant. Models including any of the points of A, points of B, or bonus (for prosocial trials only), as random slopes failed to converge due

to a sample size too small to estimate such effects for ordinal data. Judgments from Experiments 1 and 3 were used, showing no significant difference between experiments ($P > 0.27$). The last model (Experiment 1 and 3 demographics) includes self-reported demographic variables (See S9 Table) and shows a significant effect of family affluence and weekly spending money on judgments of selfish actions and of age and religion on judgments of prosocial action. Model comparison showed a significant improvement in model fits when adding the demographic variables for selfish actions (ANOVA: $P = 0.005$) but not prosocial actions ($P = 0.32$). However, the model improvement was small and did not alter the main coefficients. These statistics show that the judgments of selfish actions strongly depended on the specific situations (context, points of A and B and bonus, see Figs 2 and S3), while the judgments of prosocial actions only moderately varied across situations (S2 Fig).
(DOCX)

**S2 Table. Statistical analysis – Actions Experiment 2 and 3 main effects.** Fixed effects coefficient estimates, standard errors, and p-values of the action regressions mixed-effects models, including participants as random effects. The action data were analyzed using a binomial probit model. All continuous independent variables were normalized. Actions from Experiments 2 and 3 were analyzed, showing no significant difference between experiments ($P = 0.48$), or a marginally significant effect ($P = 0.027$) when including all demographics. The last model (Experiment 2 and 3 demographics) includes self-reported demographic variables (See S9 Table) and shows a small effect of family affluence on prosocial actions. Adding the demographic variables only marginally increased the model fit (ANOVA, $P = 0.048$) and did not alter the main coefficients. These statistics show that decisions to act selfishly strongly depend on the specific situations, in particular the cost of prosociality (bonus), consequences of selfish choices (points B), and the specific context of the decisions (Figs 2 and S3).
(DOCX)

**S3 Table. Statistical analysis – Judgments Experiment 1 and 3 efficiency and worst-off player.** Fixed effects coefficient estimates, standard errors, and p-values of the judgments regressions mixed-effects models using participants as random effects. The judgment data were analyzed using a Cumulative Link Mixed Model. Trials in which Player A picked the selfish action (took the bonus) were used for the analysis. Due to model convergence issues, only the effect of the context (Helping vs Not destroying) was included as a random slope for each participant. Models including efficiency, or score of the worst-off player, as random slopes failed to converge. Judgments from Experiments 1 and 3 were used, showing no significant difference between experiments ($P = 0.48$). These statistics show that concerns for efficiency and worst-off player affect judgments of selfish actions (Figs 3 and S3).
(DOCX)

**S4 Table. Statistical analysis - Actions Experiments 2 and 3 efficiency and worst-off player.** Fixed effects coefficient estimates, standard errors, and p-values of the action regressions mixed-effects models using participants as random effects. The action data were analyzed using a binomial probit model. All continuous independent variables were normalized. Actions from Experiments 2 and 3 were used, showing no significant difference between experiments ($P = 0.52$). These statistics show that concerns for efficiency but not for the worst-off player affect prosocial actions at the group level (Figs 3 and S3).
(DOCX)

**S5 Table. Computational models.** Utility equations and parameter description. The general utility equation for each action is described by $U(a)$ where a is the term for a generic action, $\pi_A$ and $\pi_B$ represent the generic payoffs of Players A and B, respectively. The utilities of acting prosocially ($U(P)$) or selfishly ($U(S)$) are computed for each trial, with $p_A$ and $p_B$ representing the time estimation outcomes of players A and B, respectively, and b is the bonus offered to player A. $\Delta U = U(P) - U(S)$ represents the difference in utility for prosocial versus selfish actions. All parameters are common to

the two contexts, except the bonus discount parameters, which take a different value for each context ($\delta$1 for Destroying and $\delta$2 for Helping). For parsimony, $\delta$1 was estimated relative to a $\delta$2 fixed to 1. The KW model uses data from the judgment session; $N(a)$ is the (average) judgment emitted in the judgment task for a specific action $a$. Each action $a$ is the choice taken by player A (prosocial P or selfish S), in a specific situation (payoff distributions $p_A$, $p_B$, $b$) of the action task. N(a) is computed by rescaling the judgments (entered on a scale from 1 to 6) to -1 (very socially inappropriate) and 1 (very socially appropriate) for each situation. For the original version of this model (group model), the appropriateness judgments are averaged across all participants of the judgment task of the corresponding experiment. In an individual variant of the model used in Experiment 3, we used the rating provided by each individual (i.e., estimating one's action from their own judgment in each situation).
(DOCX)

**S6 Table. Model comparison Experiments 1 and 2.** Estimated elpd (expected log pointwise predictive density) differences between all models fitted to Experiment 1, reflecting the goodness of fits of the different models using an approximate leave-one-out (LOO) cross-validation. (**a**) Models fitted to action data (SoftMax choice function) (**b**) Models fitted to judgment data (ordered-probit choice function). Model comparison shows that both actions and judgments are best explained by our extended version of the Charness and Rabin model, including a bias term and a context-specific bonus discount factor.
(DOCX)

**S7 Table. Model comparison Experiment 3.** Estimated elpd (expected log pointwise predictive density) differences between all models fitted to Experiment 3, reflecting the goodness of fits of the different models using an approximate leave-one-out (LOO) cross-validation. (**a**) Models fitted to action data (SoftMax choice function). (**b**) Models fitted to judgment data (ordered-probit choice function). The KW group models use the average appropriateness ratings of all participants of the experiment for the N(a) term, whereas the KW individual models use the participant-specific appropriateness ratings. For this experiment, participants faced only the Destroying context, the bonus discount factor capturing differences between contexts was therefore not informative, and models including this parameter were excluded for this analysis. This model comparison shows that, for Experiment 3, both actions and judgments are best explained by a trade-off model. However, the difference between the trade-off model and the version of the Charness and Rabin model including a bias term (which best explains behavior in the previous experiment) is very small.
(DOCX)

**S8 Table. Effects of the different model parameters on actions and judgments of Experiments 1–3.** Coefficient estimates, standard errors, and p-values of the mean action (**a**) or judgment (**b**) regressions fixed effect model using estimated parameters as independent variables. Average behavior and fitted parameters for each participant of Experiments 1 and 3 (judgments) and 2 and 3 (actions) are used. There are no significant differences between the two experiments for either actions ($P = 0.45$) or judgments ($P = 0.18$). These statistics show that the four parameters used in the extended version of the Charness and Rabin model (CR bias) are all relevant to explain participants' actions and judgments.
(DOCX)

**S9 Table. List of questions of the final survey on demographic and personality traits.** Demographic and personality trait questionnaires were collected: Demographic questionnaire, Altruism components of the NEO-PI-R [103] test, Levenson Self-Report Psychopathy Scale [104], Subthreshold Autism Trait Questionnaire [105], Mach-IV [106] Machiavellianism questionnaire, and 28 items of the Interpersonal Reactivity Index (IRI) [107], measuring four aspects of empathy: perspective taking (PT), fantasy (FS), empathic concern (EC), and personal distress (PD). The table reports the formulations of the questions, and the possible answers participants could choose from. Open fields indicate that participants were asked to type in their answer.

(DOCX)

**S10 Table. Statistical analysis - Experiment 4: Actions and Judgments pre-exposure.** Fixed effects coefficient estimates, standard errors, and p-values of mixed-effects regressions models of actions and judgments using participants as random effects. The action data were analyzed using a binomial probit model and judgment data using a Cumulative Link Mixed Model. Due to convergence issues, only random intercepts were used in the judgments analysis, while random slopes were included in the actions analysis. (**a**) Logistic regression of actions. (**b**) Logistic regression of actions including demographic variables. (**c**) Ordinal regression of judgments. (**d**) Ordinal regression of judgments including demographic variables. Including control demographic variables (See S9 Table) slightly improved the model fits (ANOVA $P = 0.049$ for actions and $P < 0.001$ for judgments), but did not alter the main effects coefficients. These statistics show that the behavior of the participants in the pre-exposure tasks replicates the behavior measured in Experiments 1–3, and that the choices of participants of the four exposure groups were mostly comparable (S9 Fig). There were some small differences between the pre-exposure actions of the different groups (significant effects of environment type and direction). However, they were very small in comparison to the effects of the payoffs.
(DOCX)

**S11 Table. Statistical analysis - Experiment 4: Effects of exposures to normative environments on participant's Actions.** Fixed effects coefficient estimates, standard errors, and p-values of the mixed-effects regression of actions using participants as random effects. The action data were analyzed using a binomial probit model, including random intercept and slopes for each participant, separately for each direction (a: positive and d: negative) to test for the effects of the normative environment type (prescriptive versus descriptive), and separately for the four environments (b, c, e, f) to estimate the effects of each treatment. The dummy variable Phase represents the pre- vs post-exposure choices and is significant for all environments. (**a**) Positive environments (A+ and J+). (**b**) Frequent prosocial action (A+). (**c**) Strict judgment (J+). (**d**) Negative environments (A- and J-). (**e**) Frequent selfish action. (**f**) Lenient judgment. These statistics show that participants significantly adapted their actions following exposure, and that the two types of normative environment have a significantly different effect on prosociality erosion (direction -, stronger effect of descriptive norm) but not enhancing prosocial action (direction +, no effect of the Environment type, Fig 4).
(DOCX)

**S12 Table. Statistical analysis – Experiment 4 Effects of exposures to normative environments on participant's Judgments.** Fixed effects coefficient estimates, standard errors, and p-values of the mixed-effects regression models of judgments using participants as random effects. The judgment data were analyzed using a Cumulative Link Mixed Model including a random intercept for each participant. Trials in which Player A picked the selfish action (took the bonus) were used in this analysis, separately for each direction (a: positive and d: negative) to test for the effects of the normative environment type (prescriptive versus descriptive), and separately for the four environments (b, c, e, f) to estimate the effects of each treatment. The variable Phase represents the pre- vs post-exposure choices and is significant for all environments. (**a**) Positive environments (A+ and J+). (**b**) Frequent prosocial action (A+). (**c**) Strict judgment (J+). (**d**) Negative environments (A- and J-). (**e**) Frequent selfish action. (**f**) Lenient judgment. These statistics show that participants significantly adapted their judgments following exposure, and that the two types of normative environments have a significantly different effect on judgments changes, in both positive and negative environments (S10 Fig).
(DOCX)

**S13 Table. Effects of the different normative environments and clusters on actions and judgments changes.** Analysis of variance (ANOVA) of the effects of the normative environment (Environment type and direction) as well as the types of participants (Cluster) on the average changes in judgments and prosocial actions post- versus pre-exposure for each participant. Degrees of freedom (Df), F-value, and p-value are reported for each factor. (**a**) Judgments. (**b**)

Actions. These statistics show that participants of all clusters adapted their judgments similarly following exposure, but that participants of different clusters changed their actions to a different extent after exposure to the different normative environments.
(DOCX)

**S14 Table. Effects of the different normative environments on parameter change.** Coefficient estimates, standard errors, and p-values of fixed effects regressions of fitted parameters changes (CR bias model), using as independent variables the type of environment (descriptive versus prescriptive environment types) and whether the environment was positive or negative (direction). The order of the tasks was used as a control variable. Participants' fitted parameters from experiment 4 were used. (**a**) *Baseline* preferences (Bias). (**b**) *Outcome-based* preferences ($\gamma$). (**c**) *Specific goals* ($\mu$). These statistics show that the different normative environments had different effects on the parameter changes following exposure.
(DOCX)

**S15 Table. Effects of the different clusters on parameters change post- versus pre-exposure.** Analysis of variance (ANOVA) of the effects of the types of participants (cluster), type of environment (Environment type: descriptive versus prescriptive behavior), and direction (positive versus negative changes) on the average changes in parameters of the CR bias model post- versus pre- exposure for each participant. Degrees of freedom (Df), F-value, and p-value are reported for each factor. (**a**) *Baseline* preferences (Bias). (**b**) *Outcome-based* preferences ($\gamma$). (**c**) *Specific goals* ($\mu$). These statistics showed that changes in the model parameters *outcome-based* prosocial action and *specific goals* were significantly different for distinct types of participants (cluster).
(DOCX)

**S16 Table. Overview of the four experiments.** Details of the participants (Player A) gender and age for each study, number of participants playing the role of Player B, details of the tasks and experimental sessions. See S1d and S1e Fig and S1 Text for the differences between versions.
(DOCX)

## Author contributions

**Conceptualization:** Claire Lugrin, Jie Hu, Christian C. Ruff.

**Data curation:** Claire Lugrin.

**Formal analysis:** Claire Lugrin, Jie Hu, Christian C. Ruff.

**Funding acquisition:** Christian C. Ruff.

**Investigation:** Claire Lugrin.

**Methodology:** Claire Lugrin.

**Project administration:** Christian C. Ruff.

**Supervision:** Christian C. Ruff.

**Writing – original draft:** Claire Lugrin, Jie Hu, Christian C. Ruff.

**Writing – review & editing:** Claire Lugrin, Jie Hu, Christian C. Ruff.

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
