## [Decision Letter · Decision Letter 0]

21 Nov 2024

PCOMPBIOL-D-24-01710A Computational Account of the Normative Concerns Guiding Context-dependent Prosocial Behavior

PLOS Computational Biology

Dear Dr. Hu, Thank you for submitting your manuscript to PLOS Computational Biology. After careful consideration, we feel that it has merit but does not fully meet PLOS Computational Biology's publication criteria as it currently stands. Therefore, we invite you to submit a revised version of the manuscript that addresses the points raised during the review process. Please submit your revised manuscript within 60 days Jan 21 2025 11:59PM. If you will need more time than this to complete your revisions, please reply to this message or contact the journal office at ploscompbiol@plos.org.  Please include the following items when submitting your revised manuscript: * A rebuttal letter that responds to each point raised by the editor and reviewer(s). You should upload this letter as a separate file labeled 'Response to Reviewers'. This file does not need to include responses to formatting updates and technical items listed in the 'Journal Requirements' section below. * A marked-up copy of your manuscript that highlights changes made to the original version. You should upload this as a separate file labeled 'Revised Manuscript with Track Changes'. * An unmarked version of your revised paper without tracked changes. You should upload this as a separate file labeled 'Manuscript'.If you would like to make changes to your financial disclosure, competing interests statement, or data availability statement, please make these updates within the submission form at the time of resubmission. Guidelines for resubmitting your figure files are available below the reviewer comments at the end of this letter.We look forward to receiving your revised manuscript.

Kind regards,

  Christian HilbeAcademic EditorPLOS Computational Biology  Zhaolei ZhangSection EditorPLOS Computational Biology  Feilim Mac GabhannEditor-in-ChiefPLOS Computational BiologyJason PapinEditor-in-ChiefPLOS Computational Biology**Additional Editor Comments:**  The manuscript has been evaluated by two experts. One expert has already evaluated a previous version of the paper and is fine with the current version. The other expert is generally positive but makes a number of very constructive comments to further improve the paper. I'd like the authors to take all these comments into account.  **Journal Requirements:** 

1) Please upload all main figures as separate Figure files in .tif or .eps format. For more information about how to convert and format your figure files please see our guidelines:

2) We have noticed that you have uploaded Supporting Information files, but you have not included a list of legends. Please add a full list of legends for your Supporting Information files after the references list.

3) Please amend your detailed Financial Disclosure statement. This is published with the article. It must therefore be completed in full sentences and contain the exact wording you wish to be published. Please ensure that the funders and grant numbers match between the Financial Disclosure field and the Funding Information tab in your submission form. Note that the funders must be provided in the same order in both places as well.

**Reviewers' comments:** 

Reviewer's Responses to Questions

**Comments to the Authors:**

Reviewer #1: I have reviewed an earlier version of this manuscript. The authors have addressed all my comments and I don't have any further suggestion. I think this article can be published as is.

Reviewer #2: Review for PCOMPBIOL-D-24-01710

In four behavioral experiments, the authors tested human participants in two tasks: an action task where participants decide whether to sacrifice their own payoff to help others (prosocial action) or not to help (selfish action), and a judgment task where participants judge how socially appropriate the selfish action in a specific scenario is. On the group level, participants’ actions and judgments conformed to similar social norms (Experiments 1 and 2). However, the actions and judgments of the same individual had low (Experiment 4) or even null (Experiment 3) correlations. The authors developed an adapted Charness and Rabin model to fit participants’ action and judgment behaviors, and further identified four clusters of participants based on the fitted parameters, with the clustering pattern robust across different experiments. Different types of participants exhibited different extents of behavioral changes when exposed to information on descriptive social norms (i.e., statistics of other participants’ actions or judgments).

This work has impressed me in three aspects of its findings. First, the same individual may strikingly diverge in their actions and norm judgments. Second, individuals’ actions and judgments may change with their knowledge about descriptive social norms. Third, according to an individual’s computational phenotype, the population falls into four clusters, with different types of individuals subject to the influence of descriptive social norms in different ways. I think these findings are potentially important contributions to the field, uncovering the cognitive computations behind the individual differences in prosocial behaviors. The technical quality of the work is also high. The series of experiments were carefully and cleverly designed. The data analyses and computational modeling were rigorous.

My major comments are mostly about the choice of computational models and the writing, as specified below.

1. Presentation of modeling results.

1.1. Model comparison results. In the Supplement, it says LOO was used as the metric for model comparison. But I did not find any of these results being reported in the paper, either in the main text or in the Supplement. Did I miss anything?

1.2. Model predictions versus data. Supplementary Figure S3 shows model predictions versus data for three models, but only for the main effects of single independent variables. How did the model predictions align with the interaction effects in the data, if any? Could that be visualized?

2. Alternative models to test. According to Supplementary Table S5, a total of 11 models had been tested. This looks like a comprehensive list, which includes variants of previous models. However, I noticed that some readily conceived model variants were not included. For example, why not enhance the Fehr-Schmidt inequality aversion model with an efficiency term (as several previous studies did)? Besides, the Charness and Rabin model but not the Fehr-Schmidt or the Krupka and Weber model was enhanced by both the bonus discount factor and the bias term, which seems to be unfair. Testing these additional models and comparing the performances of all models may allow the authors not only to find an even better model for human behaviors, but also to identify the specific assumptions that contribute to the success of the winning model.

3. The writing of the Introduction.

3.1. Lines 58–67: it is a little abrupt to jump from the significance of studying prosocial behaviors to “normative concerns”. What is social norm has not been defined so far. At this point in the paper, the term “normative concerns” also sounds ambiguous to me, which is reminiscent of “normative theories” in decision-making. It is introduced only later that there are two kinds of social norms: prescriptive norm and descriptive norm.

3.2. The Introduction may need rewriting to clarify the connection of the current work to previous theories and empirical findings. For example, inequality aversion can be considered as related to a specific norm – the egalitarianism norm.

3.3. In fact, the use of norm or normative concern is a little confusing throughout the paper. For example, all the terms in participants’ utility function are considered some kinds of norms. Meanwhile, what was manipulated in Experiment 4 is the knowledge of the descriptive norm, which is apparently not part of participants’ utility function.

3.4. The proposal of the four normative concerns in the Introduction may also need some rewriting. I could not understand most of them until coming to the modeling part of the Results. For example, please unpack “context-dependent preferences” using concrete examples.

3.5. The use of “harm” also puzzled me. When one decides not to help others getting rewards (in the help versus not help scenario), why should it be considered as “harming” others? Not helping does not alter the status quo. Is such use of “harm” consistent with the literature?

4. Behavioral change after exposure to descriptive norms.

4.1. The authors have shown that the different types of participants classified from model parameters differed in their change in the percentage of choosing prosocial actions. In parallel to this categorical view of the population, is there any correlation between pre-exposure parameters and the percentage change of prosocial choices? Figure 11d only shows an absence of correlation between pre-exposure parameters and the change of the same parameter.

4.2. Another interesting question is the change of participants across clusters after the exposure. A 4 by 4 matrix would be informative.

5. The across-individual correlation between judgments and actions. No across-individual correlation was found in Experiment 3 between judgments and actions, but a small yet significant correlation (r = -0.36, p < 0.001, Figure S11b) was found for Experiment 4. The lack of correlation in Experiment 3 was highlighted in the main text, while the significant correlation in Experiment 4 was hidden in the Supplement, without a word in the main text. This can be misleading. Had I not read the Supplement, I would have been convinced by Experiment 3 that the same individual’s judgments and actions have no correlation at all. The inconsistency between the two results should be disclosed and discussed in the main text.

Minor issues:

Abstract: “changing normative environments” is a little ambiguous.

Figure 4c. The percentage of participants who did not change their actions is not properly shown in the figure. It looks like a bug in figure rendering.

Figure 4d. It might be better to order the four clusters in the same order as Figure 3d.

Supplementary Table S5: How was the appropriateness judgment N(.) in the Krupka and Weber model defined?

**Have the authors made all data and (if applicable) computational code underlying the findings in their manuscript fully available?**

Reviewer #1: None

Reviewer #2: Yes

PLOS authors have the option to publish the peer review history of their article (what does this mean? ). If published, this will include your full peer review and any attached files.

**Do you want your identity to be public for this peer review?** For information about this choice, including consent withdrawal, please see our Privacy Policy .

Reviewer #1: No

Reviewer #2: No

**Figure resubmission:**  While revising your submission, please upload your figure files to the Preflight Analysis and Conversion Engine (PACE) digital diagnostic tool, https://pacev2.apexcovantage.com/. PACE helps ensure that figures meet PLOS requirements. To use PACE, you must first register as a user. Registration is free. Then, login and navigate to the UPLOAD tab, where you will find detailed instructions on how to use the tool. If you encounter any issues or have any questions when using PACE, please email PLOS at figures@plos.org. Please note that Supporting Information files do not need this step. If there are other versions of figure files still present in your submission file inventory at resubmission, please replace them with the PACE-processed versions.
---

## [Decision Letter · Decision Letter 1]

7 Apr 2025

Dear Dr. Hu,

We are pleased to inform you that your manuscript 'A Computational Account of Multiple Motives Guiding Context-dependent Prosocial Behavior' has been provisionally accepted for publication in PLOS Computational Biology.

Best regards,

Christian Hilbe

Academic Editor

PLOS Computational Biology

Zhaolei Zhang

Section Editor

PLOS Computational Biology

Originally, two reviewers have evaluated the manuscript. One of them proposed to accept the paper right away.

Now with the revised version, also the other reviewer proposes acceptance. I agree with their opinion, this is a very nice paper.

Reviewer's Responses to Questions

**Comments to the Authors:**

Reviewer #2: I have no further comments. It is a beautiful work.

**Have the authors made all data and (if applicable) computational code underlying the findings in their manuscript fully available?**

Reviewer #2: None

PLOS authors have the option to publish the peer review history of their article (what does this mean? ). If published, this will include your full peer review and any attached files.

**Do you want your identity to be public for this peer review?** For information about this choice, including consent withdrawal, please see our Privacy Policy .

Reviewer #2: No

---

## [Editor Report · Acceptance letter]

PCOMPBIOL-D-24-01710R1

A Computational Account of Multiple Motives Guiding Context-dependent Prosocial Behavior

Dear Dr Hu,

I am pleased to inform you that your manuscript has been formally accepted for publication in PLOS Computational Biology. Your manuscript is now with our production department and you will be notified of the publication date in due course.

With kind regards,

Lilla Horvath
